



# Investigation of the weather conditions during the collapse of the Morandi Bridge in Genoa on 14 August 2018

Massimiliano Burlando [1,*], Djordje Romanic [1,2], Giorgio Boni [1], Martina Lagasio [3] and Antonio Parodi [3]

[1] Department of Civil, Chemical and Environmental Engineering, University of Genoa, Genoa, Italy; massimiliano.burlando@unige.it, giorgio.boni@unige.it, dorde.romanic@unige.it, giovanni.solari@unige.it

[2] Wind Engineering, Energy and Environment (WindEEE) Research Institute, Western University, London, Ontario, Canada; dromanic@uwo.ca

[3] CIMA Research Foundation, Savona, Italy; martina.lagasio@cimafoundation.org, antonio.parodi@cimafoundation.org

*Correspondence to:* Massimiliano Burlando (massimiliano.burlando@unige.it)

**Abstract:** On 14 August 2018, Morandi Bridge in Genoa, Italy, collapsed sending vehicles and tons of rubble to the ground about 40 m below and killing 43 people. Preliminary investigations indicated poor design, questionable building practices and insufficient maintenance or a combination of these factors as a possible cause of collapse. However, at the time of collapse, a thunderstorm associated with strong winds, lightning and rain was developed over the city. While it is still not clear whether or not it played a role in this disaster, the present paper documents the weather conditions during the collapse and analyzes in detail a downburst that occurred at the time of the collapse a few kilometers from the bridge. The thunderstorm is analyzed using direct and remote measurements in an attempt to describe the evolution of the cumulonimbus cloud as it approached the coast from the sea. The detected downburst is investigated using a lidar scanner and the anemometric network in the Port of Genoa. The paper shows that the unique lidar measurements enabled a partial reconstruction of the gust front shape and displacement velocity. The Weather Research and Forecasting (WRF) simulations, carried out with three different forcing conditions, forecasted the cumuliform convection at larger scales but did not accurately replicate the downburst signature at the surface that was measured by radar, lidar, and anemometers. This result demonstrates that the localized wind conditions during the collapse time could not be operationally forecasted.

## 1. Introduction

Morandi Bridge in Genoa, Italy, named after its designer Riccardo Morandi, was built in the period 1963–1967 and it collapsed on 14 August 2018 at 11:36 a.m. local Italian time (0936 UTC). The collapse caused 43 fatalities. Morandi was known for his unconventional cable-stayed bridges that featured an unusually low span-to-stay ratio. As for the Morandi Bridge, he often used pre-stressed concrete instead of steel for the stays. Preliminary investigations after the collapse indicated poor design, questionable building practices and insufficient maintenance, or a combination of them, as possible causes of collapse. At the time of collapse, a violent thunderstorm was striking the city, but it is still not clear whether or not it played a role in this disaster due to the lack of meteorological measurements at the time of collapse in the vicinity of the bridge. Even if a quantitative evaluation of weather conditions is not available, it is evident based only on the video available very close to the accident (released by the Italian Finance Guard on July 1, 2019 and taken from a security



camera placed about 150 m to the west of the bridge), as well as many eyewitnesses, that severe weather conditions were present at the bridge around the time of collapse. Figure 1 shows two frames extracted from
the video at the time of collapse (Fig. 1a) and few seconds immediately after the bridge had fallen down (Fig. 1b). The arrow in the former picture shows the lighting that occurred to the east-southeast of the bridge during collapse, captured by the camera, while the latter picture shows the cloud of dust and debris at the ground that was caused by the bridge collapse (see Arrow 1). The finest part of the dust cloud, suspended in the air, is advected to the left of the picture by the wind that, in that moment, was blowing from south to north. Arrow 2
in Fig. 1b shows the strong inclination of a tree to the left and therefore confirms that a strong wind was likely blowing at the time of collapse. The full event recorded in a video, which is considered relevant to this study, is available at this link http://www.gdf.gov.it/stampa/ultime-notizie/anno-2019/luglio/video-del-crollo-del-viadotto-polcevera-ponte-morandi-del-14-agosto-2018#null.

For this reason, the initial speculation was that a lighting could have struck the stays or rain could have
destabilized the base thereby causing landslide. While these hypotheses were abandoned soon after, the possibility that strong winds could have played a role in the bridge collapse was considered an unlikely option by experts at the beginning of their investigations. Despite this lack of wind investigation, weather is officially ruled out as a contributing factor for the bridge collapse, at least at the time of writing. Nevertheless, a meteorological analysis of the weather conditions that occurred in Genoa at the time of bridge collapse seems
to be a relevant study in order to support this hypothesis or possibly reconsider the role of weather in this disaster.

Thunderstorms are severe weather phenomena associated with strong winds, sometimes hail, lightning and heavy rain. On average, there are 2,000 thunderstorms around the world at any given moment in time (NSSL, 2017). In mid-latitudes, thunderstorms are usually associated with low-pressure systems (e.g., cyclones) and
the Gulf of Genoa has already been identified as a region with high frequency of occurrence of cyclones (Radinović, 1987; Maheras et al., 2001; Trigo et al., 2002; Flocas et al., 2010; Kouroutzoglou et al., 2011; Romanić et al., 2016). Therefore, this region is highly susceptible to thunderstorms. Zolt et al. (2006) analyzed meteorological conditions using field observations and model simulations of the severe storm that occurred on 4 November 1966 over Italy. The investigated storm caused 118 casualties and it was associated with a cyclone
that formed in the western Mediterranean and had eastward trajectory, eventually reaching the Tyrrhenian Sea and afterwards central Italy. An overview of various severe meteorological and marine conditions associated with Mediterranean cyclones is presented in Lionello et al. (2006). However, one of the particularly dangerous weather phenomena caused by thunderstorms is a downburst.

Downburst outflows are created when a cold downdraft that originates from a thunderstorm cloud (i.e.,
cumulonimbus cloud) impinges on the surface causing radially advancing winds that are characterized with high velocities in the near-surface layer. The negative buoyancy of the downdraft is caused by evaporation, melting and/or sublimation of precipitation particles such as raindrops, hail and graupel, and additionally amplified by the drag due to the falling precipitation. Fujita (1990) reported that the wind gusts in the strong downburst outflows can reach 75 m s$^{-1}$, which is equivalent to a wind gust in an EF3-rated tornado based on
the Enhanced Fujita scale of tornadoes (McDonald et al., 2006). In addition, downburst outflows are highly transient and three-dimensional (3D), which makes them profoundly more difficult to analyze and accurately represent in space and time in comparison to the classical straight-line atmospheric boundary layer (ABL) winds. Therefore, over the last several decades a large number of research articles focused on field measurements (e.g., Fujita, 1990; Gunter and Schroeder, 2015; Gunter et al., 2017; Holmes et al., 2008; De





Gaetano et al., 2014; Burlando et al., 2018; Lombardo et al., 2018; Wang et al., 2018 Lombardo et al., 2014; Burlando et al., 2017), analytical modelling (e.g., Oseguera and Bowles, 1988; Vicroy, 1991; Holmes and Oliver, 2000; Abd-Elaal et al., 2014), physical experiments in wind simulators (e.g., Chay and Letchford, 2002; Letchford and Chay, 2002; Mason et al., 2005; Xu and Hangan, 2008; Romanic et al., 2019), numerical investigation (e.g., Straka and Anderson, 1993; Kim and Hangan, 2007; Mason et al., 2009; Orf et al., 2012)

and wind-structure interaction and wind actions (e.g., Chen and Letchford, 2004; Solari et al., 2015; Solari, 2016; Chay et al., 2006; Zhang et al., 2019; Hangan et al., 2019) of downburst outflows. This list is by no means exhaustive and there are currently (as of October 2019) almost 6,000 scientific articles with a keyword "downburst" listed on the Google Scholar portal. In addition, more than 1,000 of these articles are published since 2015, which only demonstrates the great interest and need in scientific community to better understand

the meteorological and wind engineering characteristics of thunderstorm downbursts. Indeed, a collaborative effort between meteorologists and wind engineers seem to be the most efficient way of conducting a multidisciplinary and cross-disciplinary research on thunderstorm winds with the end goal of mitigating downburst-caused damages on structures and environment (Solari, 2014; Burlando et al., 2017; Burlando et al., 2018). This article is a result of such collaboration.

Burlando et al. (2018) documented damages caused by thunderstorm downbursts in the port of Genoa on 31 August 1994. The severe winds that occurred on that day overturned more than 80% of the big cranes used to load and unload containers from cargo ships along the thunderstorm track. Shehata and his colleagues (Shehata et al., 2005) reported that the Manitoba Hydro Company in Canada lost approximately US$ 10 million in a failure of 18 transmission line towers due to the thunderstorm winds. The more well-known and, unfortunately,

more tragic accidents directly caused by downburst winds are several airplane crashes in the United States (US) (Wilson and Wakimoto, 2001). On 24 June 1975, a Boing 727 on the Flight 66 of the Eastern Airlines crashed during the landing maneuver at the John F. Kennedy International Airport in the New York City, New York, US, thereby causing 113 fatalities and multiple injuries (Fujita and Caracena, 1977). Similarly, but more recently, on 2 July 1994, a jetliner accident due to the thunderstorm downburst at the airport in Charlotte, New

Carolina, US, caused 37 fatalities and 16 injuries (National Transportation Safety Board, 1995). More downburst-related aircraft accidents are described in (Pryor, 2015). When it comes to bridges and other horizontal structures, it has been well known and vastly documented that winds can produce various along- and crosswind instability that ultimately might cause a complete structural collapse of the whole bridge or a bridge segment. Solari (2019) presents in his book a great variety of examples of wind-induced bridge collapses and

an explanation of the reasons why they happened. Further references can be found there.

    The goal of this paper is to document and analyze weather conditions prior to and during the collapse of the Morandi Bridge in Genoa on 14 August 2018. The article makes use of variety of meteorological measurements from a number of platforms such as the standard meteorological measurements from weather stations, satellite and radar observations, lightning counters, high-resolution ultrasonic anemometer measurements from the port

of Genoa as well as from a lidar scanner also installed at the port of Genoa. In addition, the field observations are complemented with numerical simulations conducted using the Weather Research and Forecasting (WRF) model with different forcing and boundary conditions, as well as the assimilated radar data. At this point, the authors want to explicitly state that this article does not suggest that the collapse of the Morandi Bridge was directly or even indirectly caused by the thunderstorm downburst or any associated thunderstorm phenomena

that occurred on 14 August 2018 in that region. However, since the exact cause and triggers to the bridge


collapse are still under investigation and largely unknown, this paper intends to provide the needed information on the weather conditions at different spatiotemporal scales prior to and during the collapse.

The rest of this paper is organized in the following manner. Section 2 describes the data and numerical simulations that were used in the analysis of weather conditions on 14 August 2018 over Genoa and the broader region. Section 3 analyzes the weather scenario prior to and during the bridge collapse with the emphasis on thunderstorm that occurred on that day over Genoa. This section analyzes the meteorological precursors for the observed thunderstorm as well as the local scale observations of thunderstorm characteristics making use of weather station data (anemometer, thermometer, etc.), Doppler radar and lidar measurements. Section 4 describes the spatiotemporal evolution of the thunderstorm using the WRF simulations and presents several derived quantitative indices used to characterize the severity of thunderstorm winds. Section 5 provides a concluding discussion on the most relevant findings presented in this study.

## 2. Data and Numerical Simulations

### 2.1. Meteorological Data

Data sources at different spatial scales have been used to describe this event from the synoptic to the local scale. The synoptic meteorological conditions have been analyzed by means of the Global Forecast System (GFS) analyses of the National Centers for Environmental Prediction (NCEP), available on a 0.5° × 0.5° grid every 6 h. The development of the cumulonimbus cloud that approached Genoa from southwest has been monitored through the meteorological radar of Liguria Region, which belongs to the Italian Meteo-radar Network and has a range of approximately 100 km. The associated lightning strikes have been recorded through the Blitzortung network (http://www.blitzortung.org/) as well as the LAMPINET system (Biron et al., 2006; Biron, 2009; De Leonibus et al., 2010), managed by the Italian Air Force. The local weather conditions have been monitored using the meteorological stations of the Meteo-Hydrological Observatory of Liguria Region, operated by the Regional Agency for Environmental Protection (ARPAL) and the weather METAR station of the Genoa Airport. Lastly, the gust front evolution in time and space has been reconstructed by means of the anemometric stations available in the Port of Genoa managed by the Port Authority, while a lidar scanner property of the University of Genoa has documented a downburst that occurred in the rear flank of the cumulonimbus cloud.

Table 1 lists the meteorological stations available in the area that was affected by the investigated thunderstorm. With the exception of GEPOA and GEPVA stations, all other stations measure precipitation (P), but only three of these stations measure wind speed and direction (W). All available measurements are either averaged or cumulated over 1 h period.

Table 2 lists the anemometric stations available within the port area. Unfortunately, wind speed measurements were available with a resolution of 1 m s$^{-1}$ only, whereas wind direction was available with 1° resolution. The sampling rate was about 4 s, which allows the evaluation of gust front movement in space and its evolution in time. However, higher sampling frequency is needed in order to precisely measure the actual maximum wind speed and turbulence intensity characteristics. Data coverage around the collapse time shows the recovery percentage of good velocity readings in a time period of 1 h centered on the bridge collapse time.

The employed lidar that has been continuously in operation since April 2018 is a 3D scanning Windcube 400S lidar developed by Leosphere. The lidar's PPI mode of operation scans the range of 100°–250° in the azimuthal direction and up to a maximum distance of 14 km. The elevation inclinations are between 2.5° and 10° from the horizontal with a 2.5° increment. Each PPI scan requires approximately 50 s. The position of the lidar is





shown in Fig. 2 (magenta triangle) together with the location of anemometer No. 11 (red square), which was installed at the lidar location. This lidar was acquired in the context of the European Project THUNDERR with the purpose of thunderstorm monitoring and detection, to measure the fine structure of gust fronts and downbursts associated with thunderstorm activity. This instrument measures the radial component of the wind

speed in the range  ±30 m s⁻¹ with a resolution of 150 m.

Figure 2 shows the position of all meteorological and anemometric stations included in Tabs.s 1 and 2, as well as the METAR station and the lidar scanner location. Note that the lidar scanner is almost side by side with anemometer 11 and thus the two symbols overlap each other. The position of Morandi Bridge is also shown in blue. The bridge was located in a suburban and industrial area along the Valley of Polcevera stream, which has

a south to north extension of approximately 10 km and riversides as high as 600 m ASL in the northernmost part and around 200 m around the area of the bridge. Anemometer 07, which is the closest one to the bridge, is 2.7 km southward from the center of the bridge, while the meteorological station GEBOL is 3.2 km northward.

## 2.2. WRF Model Setup

The Weather Research and Forecasting (WRF) model (Skamarock et al., 2008) is a compressible non-

hydrostatic model with mass-based terrain-following coordinates that was developed at the National Center for Atmospheric Research (NCAR) in collaboration with several institutes and universities for operational weather forecasting and atmospheric science research. This work adopts the WRF version 3.8.1 (Advanced Research WRF dynamic core) and the data assimilation package (WRFDA) version 3.9.1. Two different domain configurations, both operationally utilized at the CIMA (Centro Internazionale in Monitoraggio Ambientale)

Research Foundation, were implemented in the current study.

The first configuration is without any data assimilation and uses three nested domains of 13.5 (250 × 250 grid points), 4.5 (451 × 450) and 1.5 km (943 × 883) of horizontal resolution with 50 vertical levels (Fig. 3a; domains top at 50 hPa). This setup produced two different forecasts: the first numerical simulation was obtained using the initial and lateral boundary conditions from the NCEP-GFS (National Centers for Environmental Prediction

- Global Forecast System) analysis, available at a horizontal resolution of 0.25° x 0.25° and time resolution of 3 h; the second numerical simulation relied on the ECMWF-IFS (European Center for Medium-Range Weather Forecasts -Integrated Forecasting System) forcing at a horizontal resolution of 0.125° x 0.125° and 3 h time resolution. Both simulations were initialized at 0000 UTC on 14 August 2018.

The second domain setup is composed out of three nested domains of 22.5 (76 × 73 grid points), 7.5 (172 ×

163) and 2.5 km (346 × 316) of horizontal resolution and 50 vertical levels (Fig. 3b). This numerical simulation uses only the NCEP-GFS initial and boundary conditions with the additional implementation of a 3-hour cycling 3DVAR data assimilation. The assimilation was performed as follows: the WRF model was first initialized using the 0000 UTC forecasts from the GFS global model and a first 3DVAR assimilation; the model then produced a 3-h forecast followed by the second assimilation cycle at 0300 UTC and later the third assimilation

cycle at 0900 UTC; afterwards, the model generated a 24-hour forecast. In this study, the 3DVAR method was employed with a cycling update technique following the aforementioned procedure in order to assimilate radar reflectivity into the WRF model. The data assimilation was performed following the same configurations setup of Lagasio et al. (2019) and by using the modified direct operator presented in Eq. 11 in Lagasio et al. (2019).

All the simulations were performed with the same set of physical parameterizations that have already been

successfully tested in the study of similar events (Fiori et al., 2017; Lagasio et al., 2017; Lagasio et al., 2019). The surface layer was modelled using the MM5 scheme that uses the stability functions from Paulson (1970),





Dyer and Hicks (1970) and Webb (1970) to compute surface exchange coefficients for heat, moisture, and momentum. Convective velocity following the work in Beljaars (1995) was used to enhance surface fluxes of heat and moisture. The Rapid Update Cycle (RUC) scheme with a multi-level soil model (6 levels) and with a higher resolution in the upper soil layer (0, 5, 20, 40, 160, 300 cm) was used for the parameterization of land surface processes. The soil model solves the heat diffusion and Richards moisture transfer equations (with a layer approach) and in the cold season considers phase changes of soil water (Smirnova et al., 1997; Smirnova et al., 2000). The planetary boundary layer (PBL) dynamics was parameterized with the diagnostic non-local Yonsei University PBL scheme (Hong et al., 2006), which includes counter gradient terms to represent fluxes due to non-local gradients and an explicit treatment of the entrainment layer at the PBL top. The entrainment is made proportional to the surface buoyancy flux in line with results from studies using large eddy simulation models (Noh et al., 2003). The WSM6 microphysics six-class scheme was adopted to include the prediction of graupel and other microphysics processes (Hong and Lim, 2006). Finally, the radiative processes were parameterized using the longwave and shortwave RRTMG schemes (Iacono et al., 2008).

## 3. Results and discussion: observations

### 3.1. Weather Conditions at Large Scales

Between 16 and 19 August 2018, two low pressure systems named Querida and Roswitha (following the naming convention used by the Institute of Meteorology of the Freie Universität Berlin, Berlin, Germany), initially located at latitudes approximately ranging around 55°–65° N, travelled eastward following the main sub-arctic zonal flow along the storm track trough the Atlantic Ocean to the Northern Europe. On 18 August at 0000 UTC, these two depressions were situated over the northwestern Scandinavia (Querida I and II) and to the east of Iceland (Roswitha), as shown in Fig. 4a. Along with these depressions, a trough extending in a meridional direction down to the Northern Mediterranean caused instability along its path as it moved from the Bay of Biscay to the north of Italy. On the morning of 18 August, a secondary pressure minimum developed over the Gulf of Lion and moved eastward over the Ligurian Sea, bringing about an intensive thunderstorm activity. While this severe weather was particularly pronounced in the coastal areas of northwestern and central Italy, it rapidly disappeared in the afternoon. The tropopause height contour at 11,000 m (Fig. 4b; green contour) indicates that the position of the tropopause cutoff was associated with the secondary cyclogenesis that occurred in the lee of the Alps. Similar triggering mechanism of severe weather in the Gulf of Genoa was also reported by Burlando et al. (2017).

A slow time evolution of deep convective clouds above the Gulf of Genoa can be observed between 0915 UTC and 1000 UTC (Fig. 5) according to satellite images. Although the four-time instances of cloud top heights shown in Fig. 5 look similar, the noticeable feature is the connection that starts to develop between two previously disjoint cloud clusters. We observe that the clouds above the Genoa region at 0915 UTC were not connected to the cloud system that developed south from Genoa and west from Corsica. Over the next 45 min, the two regions merged to form a long convective line that stretched in the south-north direction. Around 0945 UTC, the tops of convective clouds above Genoa reached 12,000 m thereby indicating a strong and well-developed convection. These deep cumuliform clouds were responsible for the observed gust front that was detected prior to the bridge collapse (Section 3.1.3).

The radar images in Fig. 6 show the existence of two dominant precipitation cells characterized with a strong radar reflectivity (exceeding 55 dBZ). The north cell was initially situated southwest of Genoa (Fig. 8a), but in


the next 20 min it moved in the northeast direction, and between 0930 and 0940 UTC it was situated above the city. The south cell, on the other hand, stayed stationary during the same time period. One of the interesting features of the north precipitation cell is its deformation that is manifested as deeper propagation of the central
precipitation zone in the north direction. Between 0920 and 0940 UTC, the isoecho of 40 dBZ propagated approximately 6800 m inland (i.e., at 0920 UTC, this isoecho was at the coastline). Therefore, the northward propagation velocity of precipitation cell that traversed Genoa around the bridge collapse time was about 5.7 m s$^{-1}$. The northward movement of the precipitation cell is due to the orographic channeling that is inferred from Fig. 6. The mountains on both sides of two valleys that traverse Genoa rise to almost 1000 m above sea level,
while the bottom of the valley is below 100 m. The location of the Morandi Bridge in the west valley of Genoa is shown in Fig. 6 in respect to the precipitation cell. The influence of orography and river valley on thunderstorm propagation was investigated in Ćurić et al. (2003). The authors concluded that thunderstorms influenced by orographic effects are more compact in comparison to thunderstorms over a flat terrain. This difference is caused by the smaller and unidirectional (only along the direction of valley) supply of low-level
moisture in the orographic case. However, the numerical study of Ćurić et al. (2003) was carried out for a different geographic region (the Balkan Peninsula) and more research is needed on the influence of orography on gust fronts and thunderstorm propagation, in particular in coastal regions.

The 5-min Surface Rainfall Intensity (SRI) estimates from the national mosaic of the Italian Meteo-Radar Network is here used to evaluate the storm cell motion towards the coast in the period between 0900 and 1000
UTC. The application of a region growth algorithm to the binary image rain/no rain (Gonzalez and Woods, 2002) showed that the cell was isolated from the rest of the precipitation field observed at larger scales. The centre of mass of the precipitating cell was identified and the cell centroid displacement between subsequent images was used to evaluate wind speed and direction of the cell.

The results in Fig. 7 demonstrate that the precipitation cell was moving north-westward until 0925 UTC, when
the cell reached the coast. Once landed, the cell intensity decayed, its shape from approximately elliptical changed to an arrowhead-like shape pointing northward as it wedged along the Polcevera Valley and the translation speed lowered down to almost 2 m s$^{-1}$ and backed almost 90° from 0925 to 0940 UTC. The translational speed of the cell approaching the coast were on average around 4 m s$^{-1}$ with a peak speed above 6 m s$^{-1}$ between 0915–0925 UTC.

This pronounced convective activity above and south from Genoa was also detected by the Blitzortung network for lightning strikes detection (Fig. 8a). We observe that the lightning strikes were organized along a south-north line that stretched out from the north part of Corsica to north of Genoa. These results show that the lighting was predominantly concentrated over the convective line situated above the sea and the number of lightning strikes above Genoa was less significant.

The position of lightning strikes measured through LAMPINET has an accuracy of 500 m (Fig. 8b,c). In the bridge collapse hour, lightning strikes were mostly recorded over the sea (Fig. 8b) and later over the coast and east part of Genoa (Fig. 8c). The largest number of strikes occurred between 09:35 and 09:40 UTC at a distance of 4–5 km from the bridge, in the area around the Old Port of Genoa. The spatial extent of the cumuliform clouds (Fig. 5) was predominantly in the west-east, which is not necessarily the dominant direction of the area
characterized with the highest frequency of lightning strikes around Genoa (Fig. 8b,c). We find that the lightning strikes were not observed in the vicinity of the Morandi Bridge (thick black line). Therefore, the possibility that a lightning strike triggered the bridge collapse is minimal.



### 3.2. Local Observations

The standard METAR meteorological measurements from the Air Force weather station of the Genoa Airport
are shown in Fig. 9. The temporal resolution of data is not consistent throughout the records, but on average the
wind velocity and air temperature data are available at every 20–30 min, while the availability of pressure data
is at every approximately 2.5 hours. Figure 6a shows that the wind direction at the time of bridge collapse was
continuously and abruptly changing for almost 360° (i.e., from 350° to approximately 10°) in the
counterclockwise direction. At the exact time of bridge collapse, the wind was blowing from 140° (southeast).
Around the same time interval, the mean wind speed increased from approximately 2 m s$^{-1}$ to 7.7 m s$^{-1}$ at the
time of collapse (Fig. 9a). Afterwards, the wind speed dropped down to about 4.5 m s$^{-1}$ at 1000 UTC. Bearing
in mind that the values are 10-min averages, a mean velocity of ~8 m s$^{-1}$ demonstrates that the event was
characterized with relatively low wind speeds, as this value corresponds to return periods lower than 2 years in
Liguria (Zhang et al., 2018). This is further confirmed by observing the recorded wind gust of 16 m s$^{-1}$ at 0950
UTC (14 min after the collapse), which is the overall maximum value that was measured from 0940 to 0950
UTC. Note that, in the time interval between 0940 and 0950 UTC, the mean (maximum) wind speed values
measured at ARPAL stations GEPOA and GEPVA (see Fig. 2) were 7.9 (16.1) m s$^{-1}$ and 5.9 (15.5) m s$^{-1}$, and
their prevalent wind direction from south and west-southwest, respectively.

An important validation that the event was a thunderstorm downburst comes when the wind velocity records
are analyzed in conjunction with air temperature and pressure data (Fig. 9b,c). In this particular case, the air
temperature is a more useful quantity due to the higher temporal resolution of these data. Figure 9b shows that
a rapid decrease of temperature occurred simultaneously with the increase of surface wind speed and abrupt
shifts in wind direction (Fig. 9a). A sudden decrease of temperature at 0850 UTC coincides precisely with both
the increase of wind speed and abrupt change of wind direction. In the next hour (until 0950 UTC), the
temperature dropped for 4°C. This relationship between air temperature and wind velocity demonstrates that
the increase of wind speed was caused by a spread of cold air in the form of a gust front that originated from
the thunderstorm cloud (Charba, 1974; Wakimoto, 1982; Mueller and Carbone, 1987; Droegemeier and
Wilhelmson, 1987; Lompar et al., 2018). The subsequent increase of surface air pressure (Fig. 6c) is also in
accordance with the kinematics and dynamics of gust fronts, but the low temporal resolution of data prevents a
more comprehensive analysis.

The accumulated surface precipitation in the collapse hour increased in the eastward direction (Fig. 10). While
precipitation was not observed in the west parts of the port, the east side received 33.8 mm h$^{-1}$ of rain between
0900 and 1000 UTC. The weather station situated in the same valley as the Morandi Bridge (GEBOL; Fig. 10)
and approximately 3.2 km north of the bridge received 12 mm h$^{-1}$ of rain. Therefore, the zone of the strongest
precipitation coincided with the area characterized with the highest activity of lightning strikes (Fig. 8). We
also observe that the heavy rain (i.e., > 10 mm h$^{-1}$) was localized at the GEBOL, RIGHI and CFUNZ stations
(Fig. 10) thereby indicating that the Morandi Bridge probably received around 10 mm h$^{-1}$ of rain in the collapse
hour. Hail and graupel were not observed in this period at any of the weather station in the Genoa region.

Here, we discuss the high-frequency velocity records from the anemometers located along the Port of Genoa.
In the hour centered around the collapse time (0936 UTC), for the majority of stations the wind direction was
continuously changing in the counterclockwise direction starting around the north direction at the beginning of
the hour and returning back to the north direction at the end of the hour (about 1000 UTC) (Fig. 11). Although
the counterclockwise temporal change of wind direction is observed at all eight anemometers, the shift is less
pronounced at 08, 09 and 11. These three anemometers are concentrated in the west part of the port area and


approximately 5–10 km away from the other five stations. More noteworthy, the wind direction at the time of
the bridge collapse, i.e. 0936 UTC, was approximately 180° at the 07 station. At the same time, in the east
stations, i.e. 03, 01, 02, 16, the wind was blowing from the third quadrant, which is between 180° and 270°,
whereas the west stations, i.e. 08, 09, and 11 recorded the wind direction from the second quadrant, between
90° and 180°. This spatial distribution of wind directions resembles the cloud-scale (~10 km) radial outflow
produced by the gust front which was spreading from the cloud base.

Similarly, the wind speed records also exhibit noticeable differences between the west and east parts of the port
(Fig. 11). The differences between the peak wind speed close the bridge collapse time and the wind speed at
the beginning of that hour are higher at the anemometer stations located in the east part of the port (Fig. 11e–
h). The velocity time series in the west and central parts of the port (Fig. 11a–d), on the other hand, show
pronounced nonstationary signature prior to the velocity peak that occurred around the collapse time. Figure 11
shows clearly that the maximum wind speed, which occurs before 0936 UTC for stations 07, 08, 09, 11, and
after the collapse time for stations 01, 02, 03, 16, slightly shifts in time from the westernmost to the easternmost
anemometer. This can be interpreted like the signature of the gust front passage that follows the cloud translation
from south-west to north-east over approximately 15 km of coast.

Overall, these results show that the thunderstorm produced a macroburst that originated over the sea and
approached the anemometers as well as the bridge from prevailingly south direction. Recently, Burlando et al.
(2018) also demonstrated that the majority of thunderstorm downbursts in this region are generated over the sea
and then advance towards the shore. A similar trend in terms of downburst formation and movement was also
reported by Burlando et al. (2017) for a downburst event in the Port of Livorno, Italy, on 1 October 2012. As
already discussed for the METAR measurements presented in Fig. 9, the macroburst was relatively low-
intensity as the maximum velocity recorded by all stations was in the order of 15 m s⁻¹.

### 3.3. Lidar Gust Front Detection and Analysis

Approximately 15 min prior to the bridge collapse, i.e. between 0920 and 0935 UTC, the thunderstorm
downburst was also captured by the Doppler lidar installed in the Port of Genoa (Fig. 12). Note that the timing
is coherent with the maximum wind speeds recorded by anemometer 11, as shown in Fig. 11. The lidar scanned
four different elevation angles ($\varphi$) that are between 2.5° and 10° above the horizontal plane with a 2.5°
increment. These results are particularly interesting as Fig. 12 is among the first few sets of published lidar
velocity measurements of a thunderstorm downburst. Unfortunately, the heavy rain that occurred right behind
the gust front of the macroburst affected the lidar measurements reducing its range of acquisition from about 5
km at 0915 UTC to 1-2 kilometers at 0930 UTC.

This thunderstorm downburst was characterized with velocities exceeding 20 m s⁻¹ (Fig. 12). The highest
positive velocities (i.e., towards lidar) are observed at the lowest $\varphi$ and between 0915 and 0923 UTC. In
addition, Fig. 12 shows a propagation of the region characterized with the maximum velocity towards lidar
(observed for all $\varphi$ angles). The direction of the maximum velocity is from 150° until approximately 0923
UTC (Fig. 12i) and then it starts shifting in the clockwise direction eventually reaching ~170° (Fig. 12k). The
width of this region is about 3.5 km. The azimuthal orientation ($\theta$) of the zone with the maximum velocity
seems to be independent of height which indicates that the advancing gust front is not (significantly) inclined
in the $\varphi - \theta$ plane. However, it is particularly important to note that with increasing $\varphi$, this region of the
maximum velocity is inclined towards lidar. For example, when benchmarked against Fig. 12a ($\varphi = 2.5°$), this
region is closer to lidar in Fig. 12d ($\varphi = 10°$) than it is in Fig. 12e ($\varphi = 2.5°$), despite the fact that the velocity





slice in Fig. 12e is later in time than the one in Fig. 12d. The continuous advancements of this leading front between the elevations of $\varphi = 2.5°$ and $\varphi = 10°$ are nicely observed in Fig. 12b and Fig. 12c for $\varphi = 5°$ and $\varphi = 7.5°$, respectively. Moreover, the same trend is found in the next set of four velocity slices shown in Fig. 12e–h. That is, at $\varphi = 2.5°$ (Fig. 12e) the front is approximately 2000 m away from lidar, while at $\varphi = $
$10°$ (Fig. 12h), the front is already above lidar (or within 300 m away from the instrument). The intermediate distances between these two are found at the other two elevations.

This displacement of the region of the maximum velocity with height is a characteristics of the radially advancing gust front of cold air in front of the thunderstorm cloud (Charba, 1974; Wakimoto, 1982; Mueller and Carbone, 1987; Droegemeier and Wilhelmson, 1987; Lompar et al., 2018). Further, Fig. 12 enables an
estimate of the displacement velocity of the gust front towards lidar. This analysis is shown in Fig. 13 that shows the height of the gust front (black dashed line in a–g) above lidar at different radial distances from the instrument along $\theta = 151°$. For the height corresponding to 120 m ASL, (i.e., 115 m above lidar level + 5 m ASL, which is the height of the lidar ASL), we estimate the displacement velocity ($V_d$) of the gust front to be $V_{d1} = 6.3$ m s$^{-1}$ towards the instrument. A similar displacement velocity, $V_{d2} = 6.9$ m s$^{-1}$, is obtained at 70
m ASL (i.e., 65 m above lidar). Thus, an average displacement velocity is evaluated to be $V_d = 6.6$ m s$^{-1}$. Note that this velocity is about 1–2 m s$^{-1}$ larger than the estimated velocity of the north-eastward propagation of the precipitation cell in Doppler radar images (Fig. 6). The displacement velocities calculated above are very similar to the one estimated by Mueller and Carbone (1987) by tracking radar reflectivity and Doppler velocity of a gust front associated to a thunderstorm outflow measured in Denver, Colorado, in 1984. They obtained a
value of 6.9 m s$^{-1}$.

When combined with the meteorological measurements from the Genoa Airport weather station (Fig. 9) and precipitation measurements in the area (Fig. 10), this displacement velocity can be used to estimate the mean height of the cold inflow (Charba, 1974). First, we estimate the mean air densities of the ambient and gust front air masses. The air density ($\rho$) is calculated using the equation of state for wet air:

$$\rho = \frac{p_d}{R_d T} + \frac{e}{R_w T} \tag{1}$$

where $p_d$ and $e$ are the pressure of dry air and water vapor (in Pa), $T$ is the air temperature (in K), $R_d = 287.058$ J kg$^{-1}$ K$^{-1}$ is the gas constant of dry air, and $R_w = 461.495$ J kg$^{-1}$ K$^{-1}$ is the gas constant of water vapor. Since the relative humidity during the investigated time period was 100%, $e$ is equal to the saturation water vapor pressure ($e_s$) because dew point is equal to air temperature. The relationship between $e_s$ and air temperature can be expressed through the August-Roche-Magnus formula as (Lawrence, 2005):

$$e_s = e_{s0} \exp\left(\frac{A_1 t}{A_2 + t}\right) \tag{2}$$

where $e_{s0} = 610.94$ Pa, $A_1 = 7.625$, $A_2 = 243.04$℃ (Alduchov and Eskridge, 1996) and $t$ is the air temperature (in ℃). Further, the pressure of dry air is calculated using the Dalton's law of partial pressures:

$$p_d = p - e = p - e_s \tag{3}$$

since $e = e_s$ (i.e., the relative humidity of 100%). Here, $p$ is the measured (total) atmospheric pressure of wet air (in Pa). Plugging in the temperature drop of 4 K [from 296.15 K (23.2℃) to 292.15 K (19.0℃)] and the pressure rise of 130 Pa (1.3 mb) into the above equations results in the air densities of cold and warm air being
~1.2024 kg m$^{-3}$ and ~1.1833 kg m$^{-3}$, respectively.

Then, we use the equation for displacement velocity ($V_d$) of gravity currents (e.g., von Kármán, 1940; Middleton, 1966; Daly and Pracht, 1968 Charba, 1974) to estimate the mean depth of the cold inflow:



$$V_d = k \sqrt{gd \frac{\rho_2 - \rho_1}{\rho_1}} \qquad (4)$$

where $\rho_1$ and $\rho_2$ are the densities of warm (ambient) and cold (gust front) air masses (in kg m$^{-3}$), $g = 9.8053$ m s$^{-1}$ is the gravitational acceleration at 44°N and sea level, $k = 0.77$ (Seitter, 1983; Droegemeier and Wilhelmson, 1987) is the constant that represents the ratio of internal to gravitational forces (Charba, 1974), and $d$ is the unknown mean depth of the cold inflow (in m). The value of $k$ is uncertain and other figures were proposed too (Middleton, 1966; Daly and Pracht, 1968). Solving Eq. (4) for $d$ yields the value of $d = 464$ m. A value of 478 m is obtained if the air is assumed to be dry. For instance, Charba (1974) used Eq. (4) to estimate the displacement velocity of 22.7 m s$^{-1}$, but the measured value of $d$ in his study was 1350 m. Smaller values of $V_d$ and $d$ were reported in Goff (1976). Besides the value of $k$, the largest uncertainty associated with the above analysis is related to the usage of surface air densities in Eq. (4) instead of the mean air density along the height of gust front inflow. Unfortunately, temperature and pressure profiles at this location are not available for this event. Since the air density difference $\rho_2 - \rho_1$ is fairly constant with height (Charba, 1974) and $\rho_1(z > 0) < \rho_1(z = 0)$, the obtained value represents the upper limit of $d$. The higher wind speeds associated with the gust front passage (Fig. 11 and Fig. 12) in comparison to displacement velocity of the gust front (Fig. 13) are expected and reported elsewhere too (Charba, 1974). According to Goff (1976), the slowly moving gust fronts such as the one reported in this present paper, are usually associated either with intensifying storms and accelerating outflow or with dissipating storms and decelerating outflows with respect to the storm. On the other hand, gust front velocities are usually the highest during the mature stage of a thunderstorm cloud.

The gust front leading edge was inclined in the direction of propagation (i.e., towards lidar) due to the increase of wind speed with height in the outflow (dashed line in Fig. 13). The angle of inclination depends on density differences between ambient and gust front air masses and wind speed, among other factors. In the analyzed case, the inclination angle is -10.1° from the horizontal plane, with the minus sign indicating the negative inclination in respect to the measurement location. That is, the gust front surge line increases for 177 m km$^{-1}$, which is very similar to the result of Charba (1974), who obtained the value of 150 m km$^{-1}$ for a gust front that occurred in central Oklahoma, United States. This comparative analysis indicates that the surface roughness might not be the dominant factor that governs the slope of the gust front leading edge (i.e.., gust front surge line) because similar values are obtained for two profoundly different surface roughnesses. However, this subject deserves more research and larger sample of data in order to confirm this speculation.

Further reconstruction of this event is possible by applying various relationships between the height of cold inflow ($d$) and other features of the gust front. According to a number of studies that investigated gust fronts at full scale or using physical experiments and numerical simulations (e.g., Goff, 1976; Charba, 1974; (Wakimoto, 1982; Droegemeier and Wilhelmson, 1987), the height of the leading edge ($H$; called gust front head) is usually $H \cong 2d$. This further enables us to sketch the likely shape of the gust front that occurred over the sea approximately 15 min prior to the bridge collapse (Fig. 14). The characteristic heights are indicated in figure with the height of 159 m being the first lidar slice at $\varphi = 10°$ (see Fig. 13). The height of the gust front nose (265 m) is estimated from Charba (1974), where the author found that this height is approximately $d/1.8$. However, other relationships are reported in Goff (1976); hence there is uncertainty associated with this value. Note also that the height of the gust front nose is not the height of maximum horizontal wind speed which is however much lower and closer to the ground, as also commented before according to the fact that the highest velocities in Fig. 12 are observed for lidar scans with $\varphi = 2.5°$. The slope of ~70° of the gust front edge above the height of the nose is adapted from Goff (1976), whereas the gust front head height (928 m) is estimated





through the relationship $H \cong 2d$. The region behind the gust front head is characterized by pronounced turbulence due to the Kelvin-Helmholtz instability (Britter and Simpson, 1978) and therefore there is no a clear shape of the gust front edge in that region (notice that the blue dotted lines are not connected behind the gust front head in Fig. 14).

In consistency with the gust front and downburst outflow theory (Charba, 1974; Droegemeier and Wilhelmson, 1987; Mueller and Carbone, 1987; Lompar et al., 2018), Fig. 12c,d,f,g,h shows that the gust front was associated with two high speed regions that were separate with a zone of smaller wind speeds. This pattern is particularly noticeable at higher elevations (Fig. 12c,d,g,h). The above-referenced studies demonstrated that the cold inflow typically comes in periodic surges, which is further schematically depicted in Fig. 14. The separation between two high-speed surges increases with height and decreases with time. For instance, the distance between these two regions in Fig. 12c ($\varphi = 7.5°$) is approximately 470 m and it increases to around 610 m in Fig. 12d ($\varphi = 10°$). However, for the same scanning angle, but later in time (Fig. 12h), the second high-speed surge almost connected with the leading edge of the gust front. These periodic surges are absent at the lowest $\varphi$.

To conclude the discussion on lidar data, Fig. 15 shows the lidar velocities around the collapse time. The gust front that was clearly visible in Fig. 12 has passed the lidar location and only small velocities (below 10 m s$^{-1}$) were recorded around the bridge collapse time. A positive (towards lidar) and negative (away from lidar) velocity regions were detected for $\theta < 150°$ and $\theta > 200°$, respectively, at all four elevation angles. Wind speed was increasing with height, but overall there are no pronounced differences between different heights. Around the collapse time (~0936 UTC), the core of precipitation cell in Fig. 6b,c had already passed over lidar and therefore the negative Doppler velocities for $\theta > 200°$ are observed. The radial flow field measured by the lidar in Fig. 15 indicates that the winds were characterized with westward direction (east to west).

## 4. Results and discussion: WRF Numerical Simulations

The WRF numerical simulations results have been firstly analyzed by computing the 5-minute maximum wind speeds at 10 m above ground and at the location of the eight anemometers in the Port of Genoa. The considered period is 0600–1500 UTC. The WRF-IFS, WRF-GFS and WRF-GFS-DA predictions have been interpolated at the anemometer locations employing the nearest neighbor interpolation method. The results in Fig. 16 show that WRF-IFS experiment is capable to predict a maximum wind speed up to 11 m s$^{-1}$ in the period between 0800 and 0900 UTC, whereas it predicts a higher maximum between 1200 and 1300 UTC. However, the WRF-GFS and WRF-GFS-DA simulations produce significantly lower values of 5-minute maximum wind speed at all eight locations (Fig. 16).

To gain a deeper understanding of this finding, the vertical maximum intensity (VMI) maps of radar reflectivity from these three different forcing conditions were numerically investigated at the time of occurrence of the wind speed maxima mentioned above. The first two peaks are assumed to occur roughly at the same time for all the three WRF configurations, at 0830 and 0955 UTC (Fig. 17). The last peak, which is also the most intense one, occurs at 1240 UTC in the WRF-IFS simulation, at 1325 UTC in WRF-GFS, and at 1310 UTC in WRF-GFS-DA. For comparison, Fig. 17(a-c) shows the VMI measurements at 0830, 0955, and 1325 UTC. The VMI observed at 1240 and 1310 UTC are not reported as the corresponding patterns of reflectivity are analogous to the one at 1325 UTC (Fig. 17c). In general, one can observe that there is never correspondence between concurrent reflectivity patterns of simulations and observations, which is not much surprising as numerical weather prediction models very often present time and space shift between forecast and measurements,





especially for small-scale phenomena like single cell thunderstorms. Some deep convective spots, with VMI values above 40 dBZ, occur in all the three WRF simulations at 0830 and 0955 (Fig. 17d,g,j and e,h,k), but these structures are rather small, organized along narrow stripes from southwest to northeast, and they do not resemble the structures in Fig. 17a,b whereas they are more similar to the convection that develops later after noon (Fig. 17c at 1325 UTC,). WRF-IFS at 1240 UTC seems to be able to produce a convective storm with better spatial agreement with the one observed four hours before at 0830 UTC (Fig. 17a), producing wind speeds as high as 16 m s$^{-1}$, which is the same maximum value observed in anemometric measurements at the collapse time (Fig. 11).

According to the latter consideration, we assume that WRF-IFS at 1240 UTC is the most reliable representation of the thermodynamic conditions to produce strong single cell thunderstorms similar to the one observed during bridge collapse. Hence, the corresponding skew$T$-log$p$ diagram has been computed at position (Lon 8.8° E, Lat 44.4° N) in front of the Genoa harbor (Fig. 18). The diagram presents a significant MUCAPE (Most Unstable Convective Available Potential Energy) equal to 2934 J kg$^{-1}$, which is even higher at 0830 UTC (3427 J kg$^{-1}$) and 0955 UTC (3678 J kg$^{-1}$) despite no strong convective cells are detected earlier in the morning. Precipitable water is 30 mm for the WRF-IFS experiment, thus corresponding to a scenario prone to the occurrence of high precipitation. This figure is similar to the amount of rain measured at the CFUNZ station in the east part of Genoa (Fig. 10). Considering the SWEAT and Showalter Index (SWI) values, it seems that in the WRF-IFS simulation both indicate a low probability for severe thunderstorm to occur. Similar conclusions can be drawn when considering the Total Totals as well as the K and Lifted Index values. Wind shear and storm relative helicity (SREH) depict atmospheric conditions that are not prone to develop supercell-like structures. Finally, it is worth highlighting that the value of the Microburst Windspeed Potential Index (MWPI), which by definition ranges between 0 and 5, here is equal to a value of 3.2 that corresponds to thermodynamic conditions prone to the occurrence of downbursts (Pryor, 2015).

All considered, the current WRF simulations with three different forcing conditions were not capable of fully replicating the transient wind characteristics in the bridge collapse hour. WRF-GFS simulations were not able to produce a maximum wind speed of approximately 16 m s$^{-1}$, which was the wind gust measured at the nearby station of Genova Airport as well as the other stations in the port area around the collapse time (Fig. 9), while WRF-IFS produced a thunderstorm cell similar, in terms of intensity of wind speed and reflectivity, to the one that occurred around 0930 UTC, roughly at the same location but a few hours later. This demonstrate that the WRF forecast of radar reflectivity, thermodynamic properties of the atmosphere at larger scales, and wind field at the local scale can be used potentially as a useful tool for determining the meteorological precursors and contributing factors to the development of thunderstorms in this region. Also, in perspective these simulations might be used to simulate gust fronts and downbursts at much finer scale adopting proper downscaling techniques or coupling WRF with other higher resolution models (Chen et al., 2011; Solari et al., 2012; Burlando et al., 2007).

## 5. Conclusions

This article documents and analyzes the weather conditions prior to and during the collapse of the Morandi Bridge on 14 August 2018. This disastrous event that occurred at 0936 UTC caused 43 casualties. Since the forensic investigation of collapsed bridge has not pinpoint the exact factors that caused the collapse, the goal of this paper is to provide the contributions in terms of observed and modelled weather conditions around the

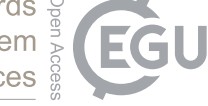
collapse time. Therefore, the analyses in this study are based on direct meteorological measurements from the weather stations in the area, wind velocity records from eight anemometers installed along the Port of Genoa, remote satellite and radar observations and Global Forecast System (GFS) analysis. The local weather station of Genoa Airport recorded the wind gust of 16 m s$^{-1}$ few minutes after the collapse time. The time records of air temperature, pressure and wind velocity from this weather station, as well as the higher frequency velocity measurements from the eight anemometers located along the coastline showed the well-known signature of transient thunderstorm conditions during the collapse hour. This observation was confirmed using the satellite (i.e. cloud top heights) and radar (i.e., composite radar reflectivity) data. The radar data showed the existence of two cells close to Genoa with the strong radar reflectivity exceeding 55 dBZ. The satellite images, on the other hand, depicted a convective line that stretched in the south-north direction and extended from Corsica to norther Italy (thus crossing Genoa). The event was also characterized with the high frequency of lightning strikes, but they were not recorded exactly in the close vicinity of the Morandi Bridge but a few kilometers east of it.

In addition, this study presents the unique lidar measurements of downburst outflow in the form of gust front that advanced in ahead of the parent cumulonimbus cloud. Furthermore, we developed and applied a technique for gust front surge reconstruction using lidar measurements and theoretical relationships that analytically link the main parameters of this phenomena. The displacement velocity of gust front that was measured approximately 20 min prior to the bridge collapse was estimated to be 6.6 m s$^{-1}$ and the height of the cold pool behind the gust front head was calculated to be at 464 m. The front was advancing from the sea towards the land and the observed features were measured approximately 10–15 km south from the bridge. In addition, the lidar velocities showed multiple high velocity spots behind the leading edge of the gust front.

The numerical simulations were carried out using the Weather Research and Forecasting (WRF) model with three different initial and boundary conditions: (1) IFS (from the European Center for Medium-Range Weather Forecast); (2) GFS (from the National Centers for Environmental Prediction) and (3) GFS-DA. The GFS-DA simulation uses the assimilated radar reflectivity data into the GFS analysis. While all three numerical simulations under-estimated the maximum velocity during the bridge collapse hour, the simulated radar reflectivity and thermodynamic indices demonstrated that these products can be used as precursor alerts for severe thunderstorm weather in this region.

In conclusions, this paper demonstrated the existence of a severe thunderstorm in the Genoa region during the collapse of the Morandi Bridge on 14 August 2018. However, while the strong and non-synoptic wind conditions in the form of thunderstorm downburst and gust front, as well as precipitation and lightning, were all present in the minutes prior to and during the bridge collapse, this study does not firmly conclude that the severe weather was the major triggering factor for the collapse. However, we do acknowledge that the documented weather conditions might have played some role in the failure of the Morandi Bridge.

**Author contributions**

MB supervised and designed the study, and was responsible for data collection. MB and DR performed data analysis and manuscript preparation and editing. DR performed literature review. ML and AP performed numerical simulations, analyzed numerical results and edited the corresponding sections in this paper. GB analyzed radar reflectivity and storm motion.



**Competing interests**

The authors declare that they have no conflict of interest.

**Acknowledgments**

The contribution of Massimiliano Burlando and Djordje Romanic to this research is funded by the European
Research Council under the European Union's Horizon 2020 research and innovation program (grant agreement
No. 741273) for the project THUNDERR - Detection, simulation, modelling and loading of thunderstorm
outflows to design wind safer and cost-efficient structures – through an Advanced Grant 2016. The authors
thankfully acknowledge the cooperation of the Port Authority of Genoa for providing the anemometric
measurements in the port of Genoa. Satellite images are based on level 1 data recorded by SEVIRI instrument
on board Meteosat Second Generation satellites, operated by EUMETSAT. Thanks are due to LRZ
Supercomputing Centre, Garching, Germany, 845 where the numerical simulations were performed on the
SuperMUC Petascale System, Project-ID: pr62ve.

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

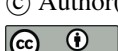


**Table 1. Meteorological stations managed by the Regional Agency for Environmental Protection of Liguria Region available in the coastal area affected by the thunderstorm on 14 August 2018.**

| Station ID | Coordinates (lon °E, lat °N, altitude m ASL) | Parameters[1] |
|---|---|---|
| GEBOL | (8.89561, 44.45530, 47) | T, PR |
| RIGHI | (8.93433, 44.42797, 360) | T, PR, W |
| CFUNZ | (8.94591, 44.40035, 30) | P, T, R, PR, RAD |
| GEPCA | (8.96109, 44.43439, 30) | PR |
| GEPEG | (8.82460, 44.43227, 69) | T, PR |
| GEQUE | (8.97260, 44.42367, 200) | T, RH, PR |
| MADGR | (8.74299, 44.43344, 104) | T, RH, PR |
| MGAZZ | (8.84485, 44.44247, 310) | T, PR |
| GEPOA | (8.92317, 44.40816, 25) | W |
| GEPVA | (8.95222, 44.39278, 10) | W |

[1] Parameters are P = pressure (hPa), T = temperature (°C), RH = relative humidity (%), PR = cumulated precipitation (mm/h), W = wind speed (m/s) and direction (°), RAD = solar radiation (W m$^{-2}$)




**Table 2. Specification of the anemometric stations managed by the Port Authority of Genoa.**

| Station ID | Coordinates (lon °E, lat °N, altitude m ASL) | Data coverage 14 Aug 2018 | Data coverage 0900-1000 UTC |
|---|---|---|---|
| 01 | (8.91838, 44.41119, 16) | 66.6 | 78.8 |
| 02 | (8.91939, 44.40780, 17) | 64.9 | 81.6 |
| 03 | (8.91501, 44.40771, 15) | 72.4 | 79.8 |
| 07 | (8.88499, 44.40220, 15) | 84.4 | 85.7 |
| 08 | (8.83541, 44.41562, 16) | 66.5 | 90.8 |
| 09 | (8.82859, 44.41862, 15) | 65.9 | 87.9 |
| 11 | (8.77698, 44.41754, 25) | 62.7 | 88.0 |
| 16 | (8.92797, 44.40157, 18) | 72.4 | 78.0 |




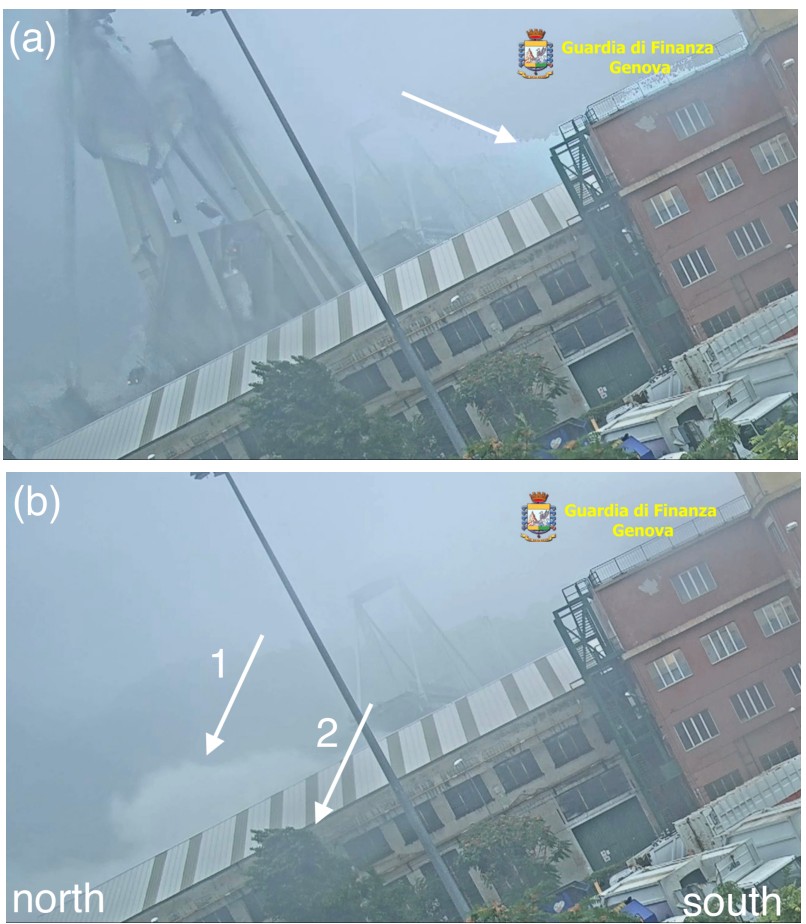

Figure 1. Two frames of the video recorded by a security camera located nearby and to the west of the bridge taken during the collapse of Morandi Bridge (a) and immediately after (b).
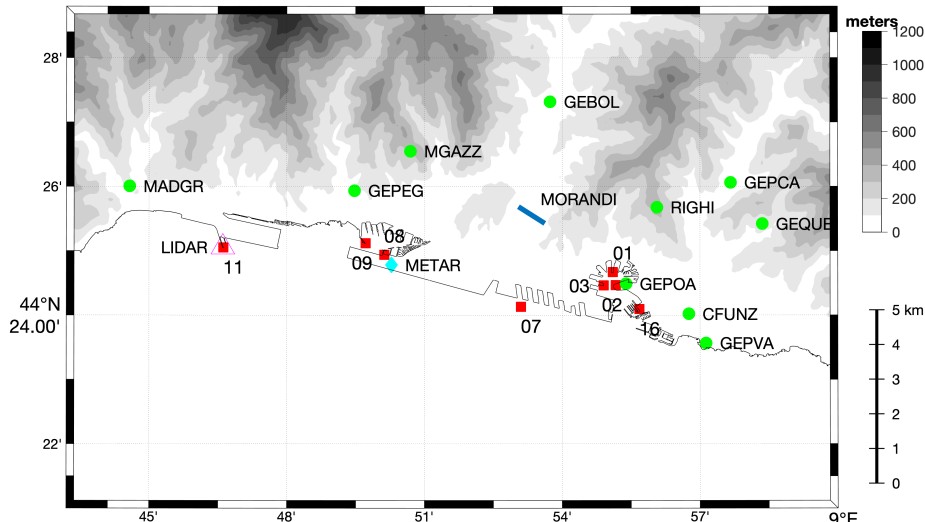

**Figure 2. Topographic map of the Genoa region showing the position of Morandi Bridge (blue line), ARPAL meteorological stations (green circles), Port Authority anemometric stations (red squares), airport METAR station (cyan diamond), and lidar scanner of the University of Genoa (magenta triangle).**




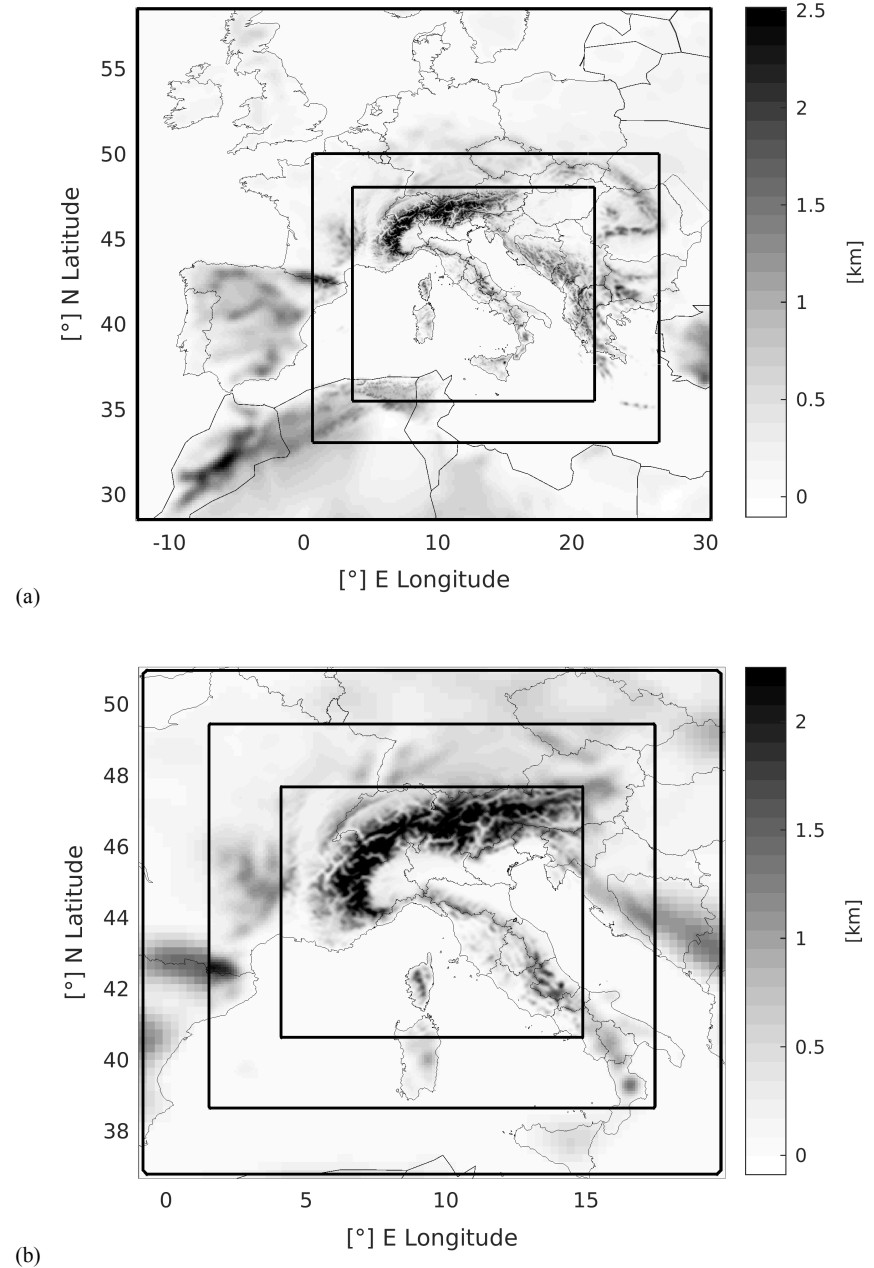

**Figure 3. WRF domains extension used for the simulations without (a) and with (b) data assimilation.**
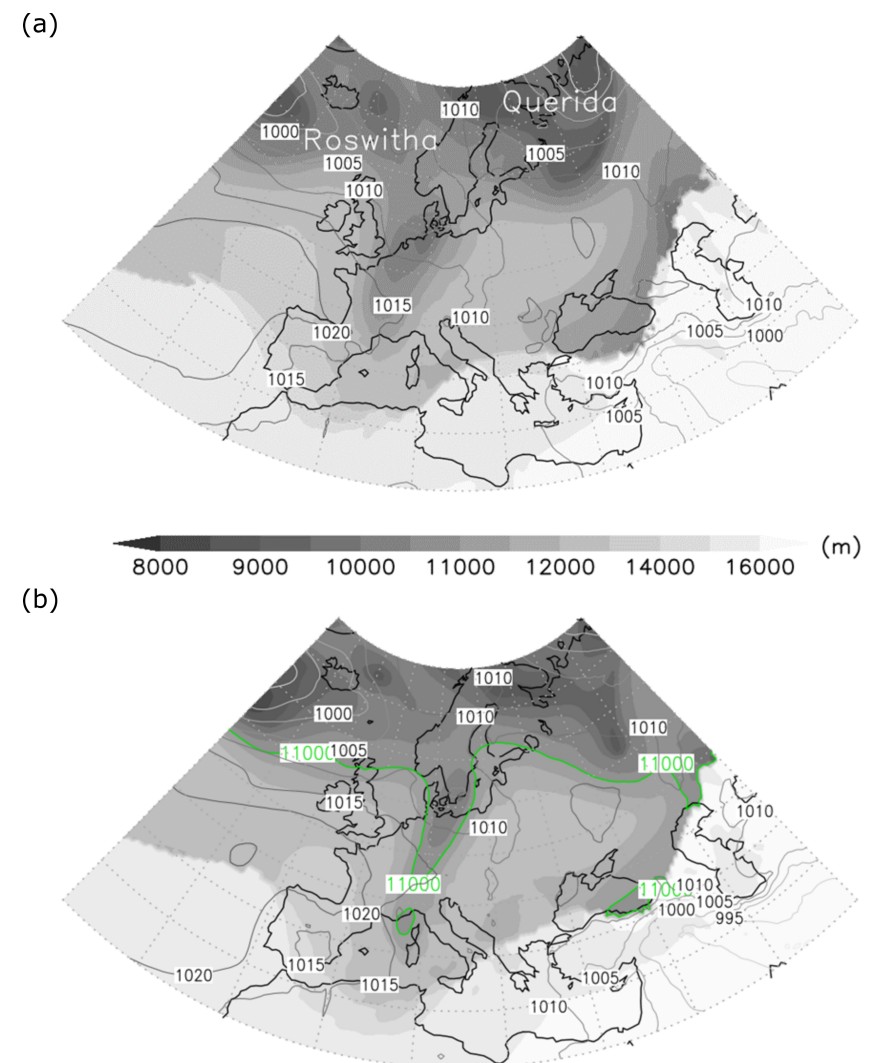

**Figure 4. Mean sea level pressure (contours) and tropopause height (shaded contours) over Europe from GFS analyses (a) at 0000 UTC and (b) at 1200 UTC. The green contour shown in the panel below corresponds to the cutoff that occurred over the Ligurian Sea in the morning on 14 August 2018.**



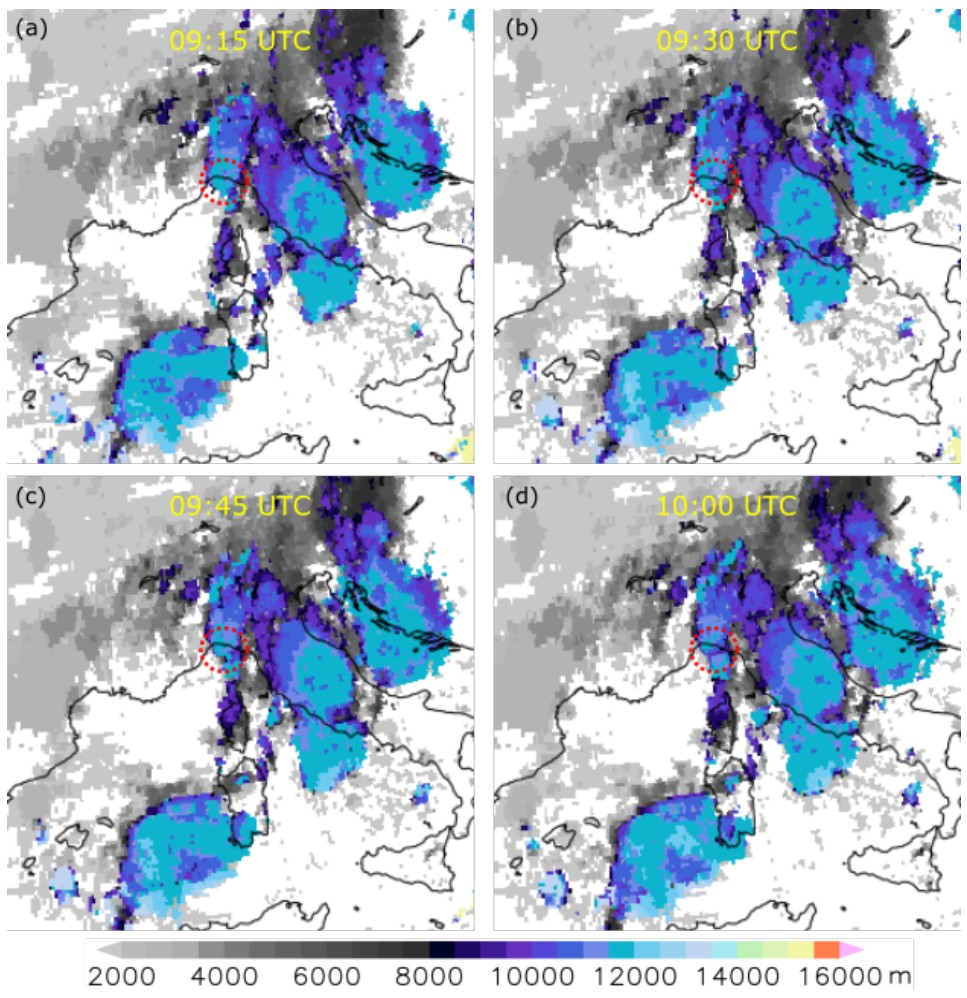

**Figure 5. Cloud top heights above Italy and surrounding regions from the MSG (Meteosat Second Generation) satellite data. The retrieval time of images is indicated in each plot and the red (dashed) circle represents a wider region around Genoa.**




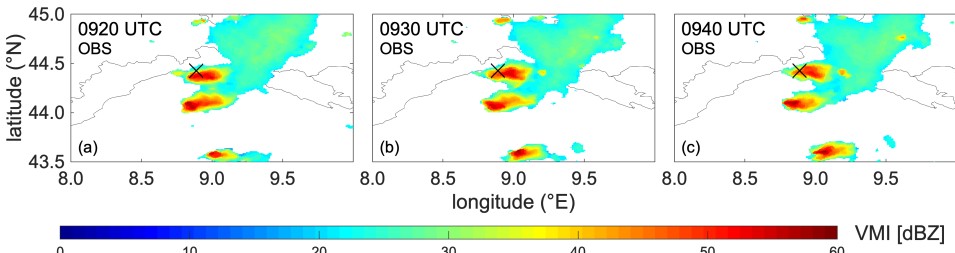

**Figure 6. Radar reflectivity (dBZ) measured by the Ligurian Doppler radar. The panels (a) to (c) show three consecutive VMI (Vertical Maximum Intensity) images at 0920, 0930, and 0940 UTC, respectively. The position of Morandi bridge is indicated by the black cross.**



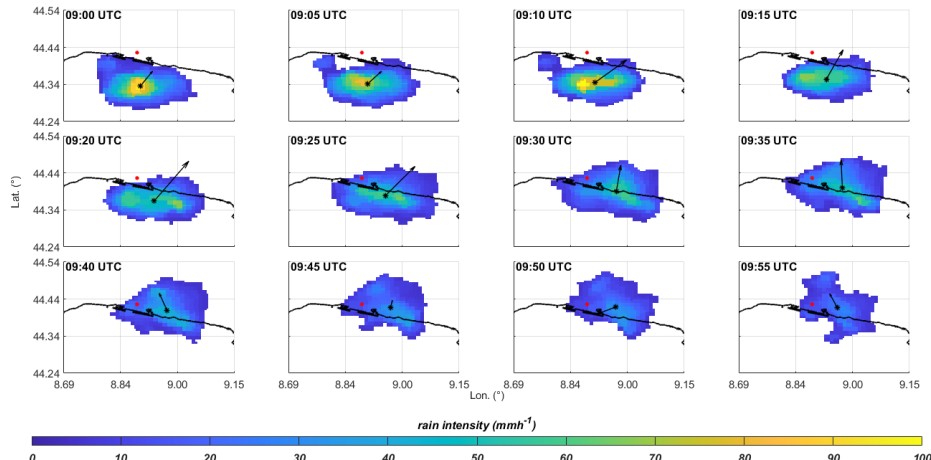

**Figure 7. Rain cell observed offshore the Port of Genova on 4 August 2018 between 0900 and 0955 UTC. The arrow is placed in the cell's center of mass and its length is proportional to cell's translational velocity. Observed 5-min mean intensities were as high as 100 mm h⁻¹.**


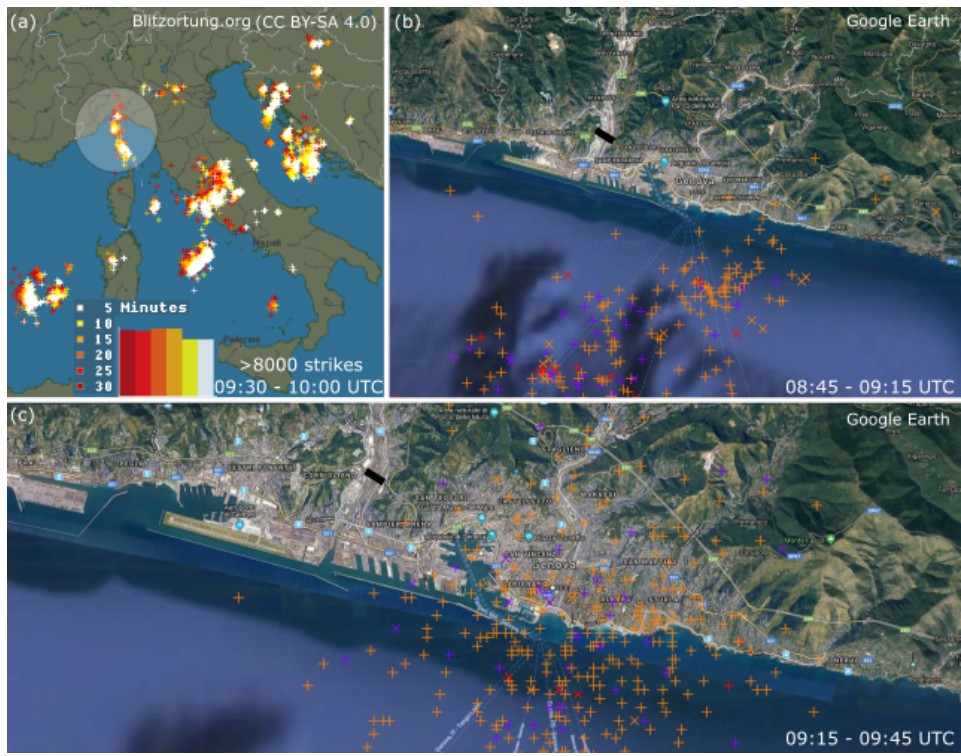

**Figure 8. (a)Lightning strikes recorded between 0930 and 10:00 UTC on 14 August 2018 over Italy and the surrounding regions (Source: Blitzortung.org). (b,c) LAMPINET counts of lightning strikes above Genoa in two different time intervals (Background maps source "© Google Earth"). The thick black line in (b) and (c) shows the location of Morandi Bridge.**

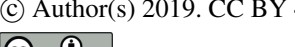


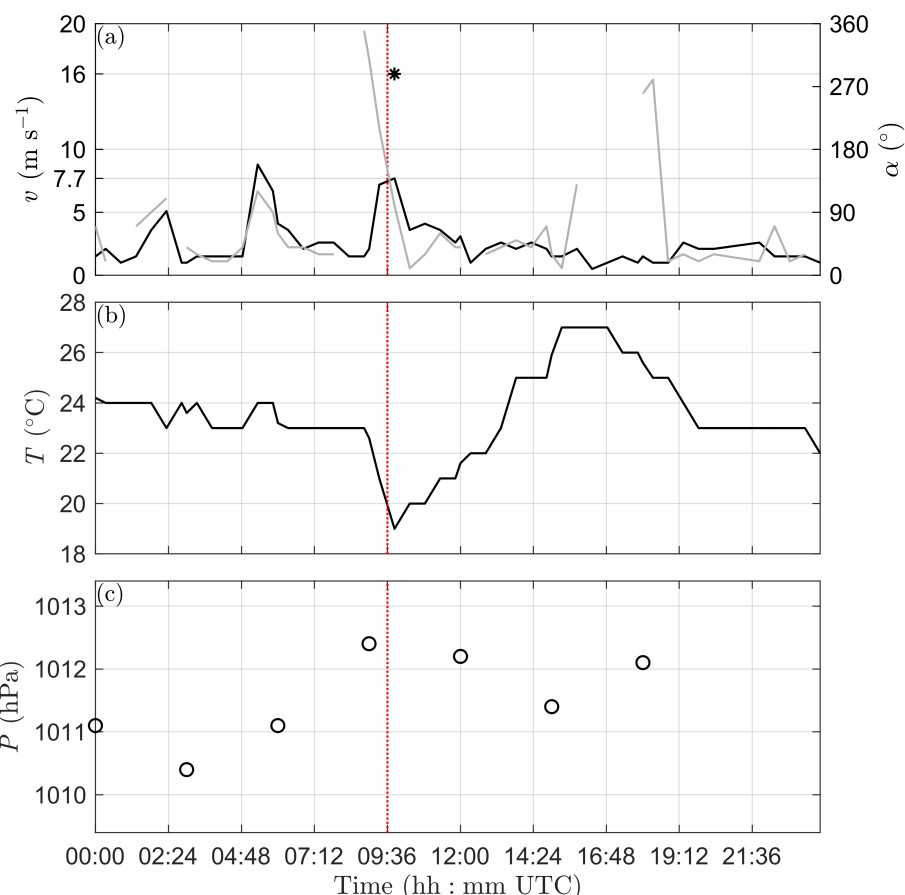


**Figure 9. Meteorological measurements from the Genoa Airport weather station on 14 August 2018: (a) wind speed (black line; primary *y*-axis) and wind direction (grey line; secondary *y*-axis); (b) air temperature at 2 m above ground; and (c) sea level pressure at 2 m above ground. In (a), the asterisk symbol shows the velocity peak. The vertical red (dotted) lines indicate the bridge collapse time.**



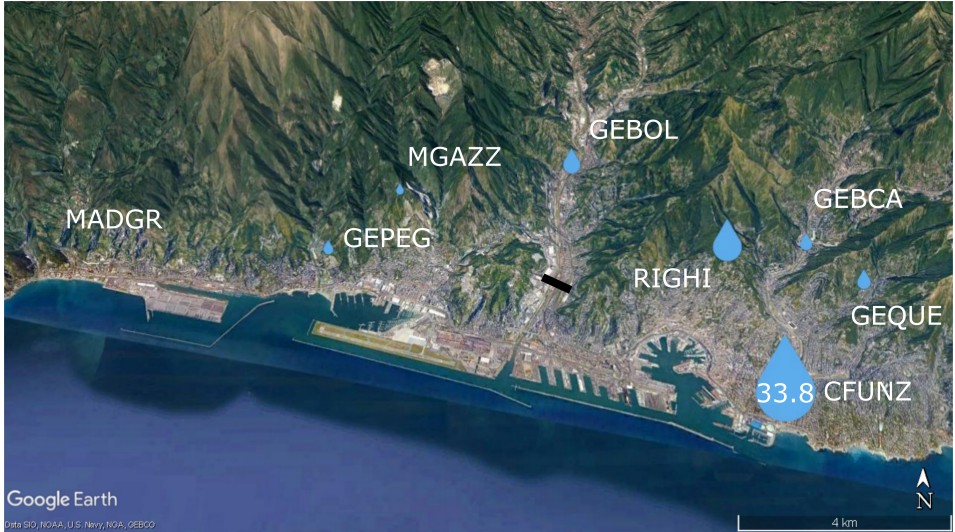

**Figure 10. Accumulated surface precipitation between 0900 and 1000 UTC in Genoa. The accumulated precipitation at the CFUNZ station shown in mm h–1 and the size of other droplets scaled accordingly. Background map source "© Google Earth".**




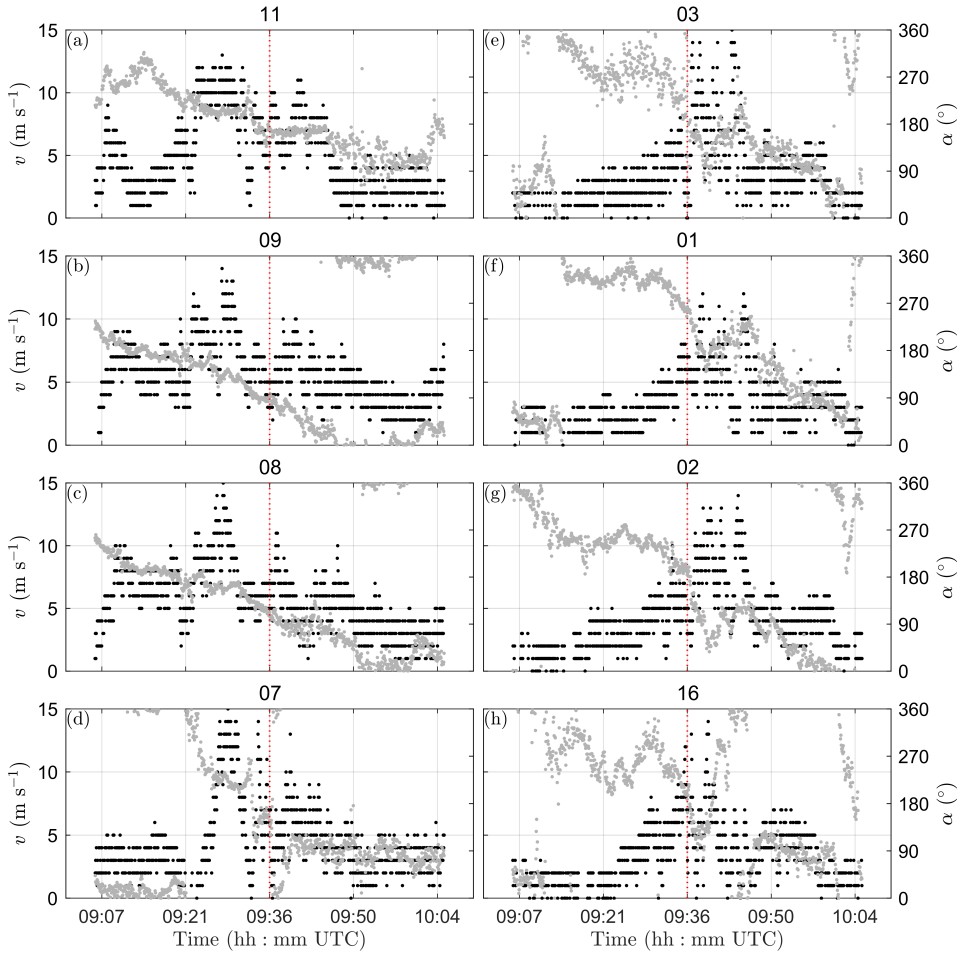

**Figure 11. Wind speed (black dots; primary $y$-axis) and wind direction (grey dots; secondary $y$-axis) from 8 anemometers in the Port of Genoa. The vertical red (dotted) lines indicate the bridge collapse time. Panels (a)–(d) represent the stations west and south from the Morandi Bridge, while the panels (e)–(g) are the stations southeast from the bridge (see Fig. 2 for details).**


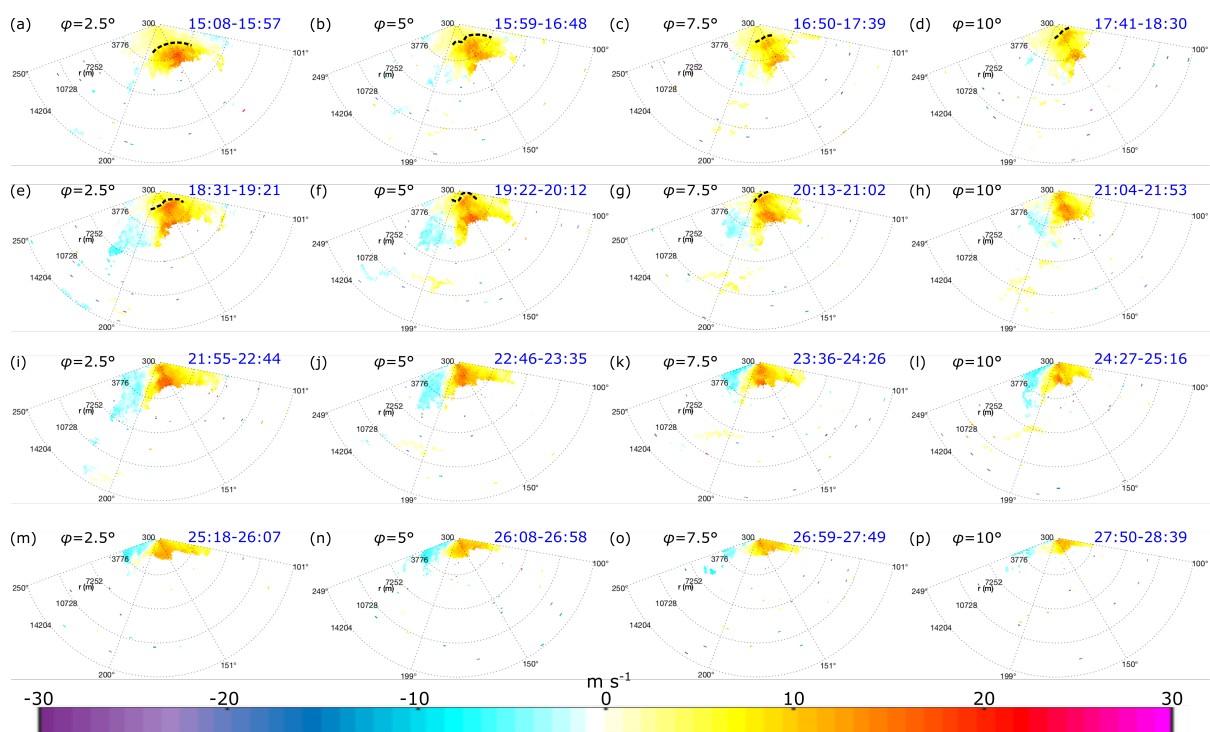

**Figure 12.** Doppler velocity measured by the WSL400s lidar located in the Port of Genoa. The beam elevation ($\varphi$) and the scanning time interval in MM:SS (blue text; the hour in all plots is UTC) provided in each figure. The black dashed line in (a) to (g) shows the leading edge of the maximum velocity region.

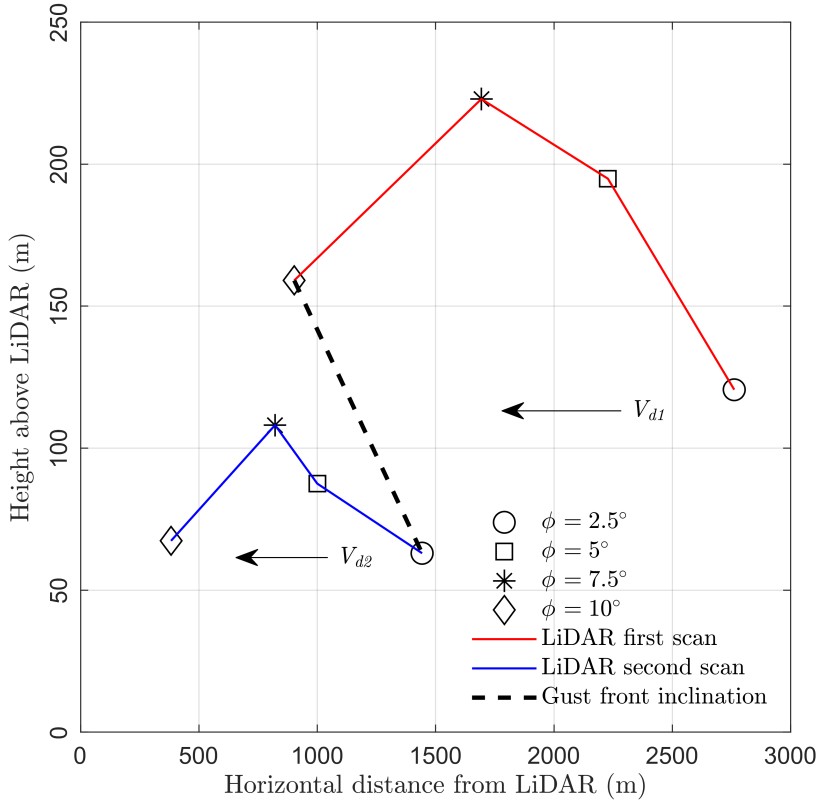

**Figure 13. Post-processed lidar data shown the height of gust front at different elevation angles (symbols) and different scanning times (colored lines). Two black arrows show the estimated value of displacement velocity at 115 m ($V_{d1}$) and 65 m ($V_{d2}$) above lidar. The black dotted line depicts the inclination of the gust front head.**


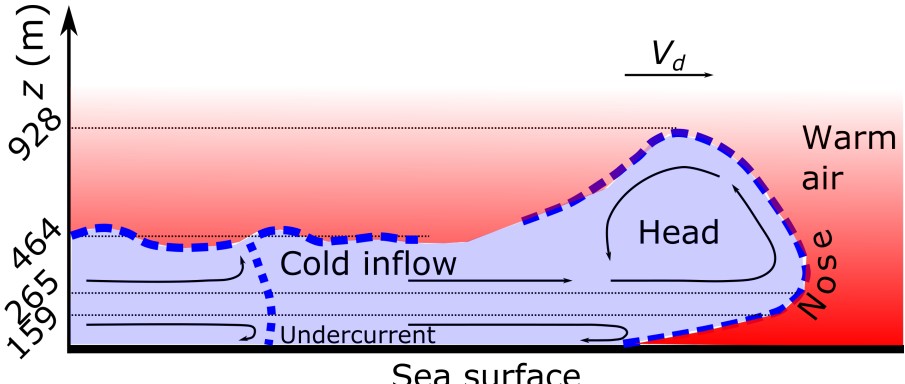

**Figure 14. Conceptual model of the gust front that occurred prior to the collapse of the Morandi Bridge in Genoa on**
       **14 August 2018. The blue dashed line represents the estimated shape of the gust front. The blue and red colors**
       **indicate the regions of cold and warm air, while the black $V_d$ arrow shows the direction of gust front propagation.**
       **The black arrows within the gust front portray the air movement relative to the front. See text for full interpretation**
       **and underlying assumptions.**






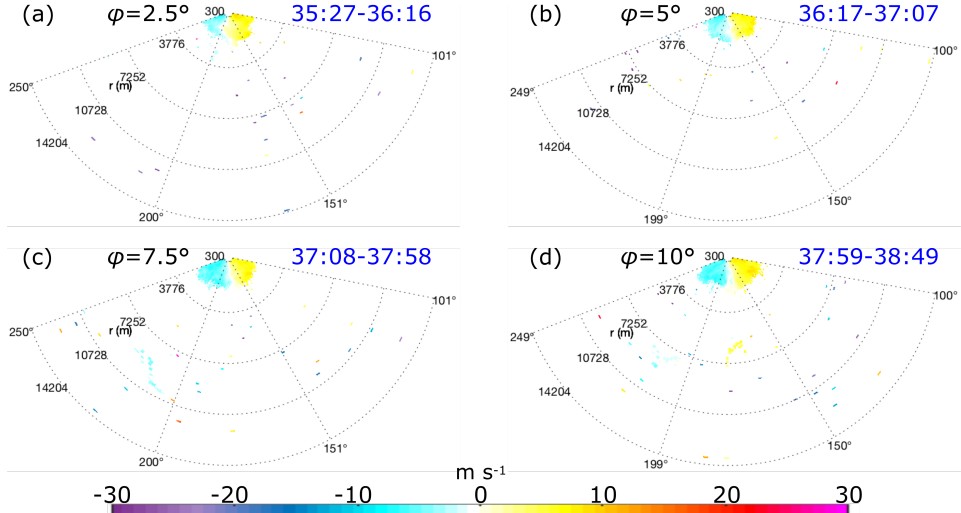

**Figure 15. Doppler velocity measured during the time of bridge collapse by the WSL400s lidar located in the Port of Genoa. The beam elevation (φ) and the scanning time interval in MM:SS (blue text; the hour in all plots is 09 UTC) provided in each figure.**






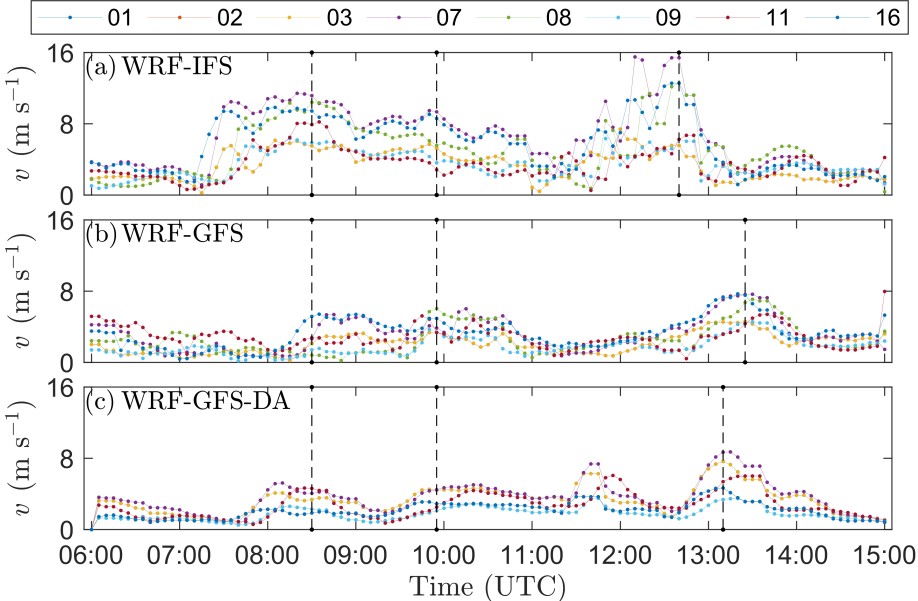

**Figure 16. Maximum 5-min wind speeds at 10 m above ground at the eight anemometers location in the Genoa port (see Fig. 2 and Fig. 11 for anemometer locations and measurements, respectively). The vertical dashed lines show the time instances used for the analysis of radar reflectivity and thermodynamics diagrams.**






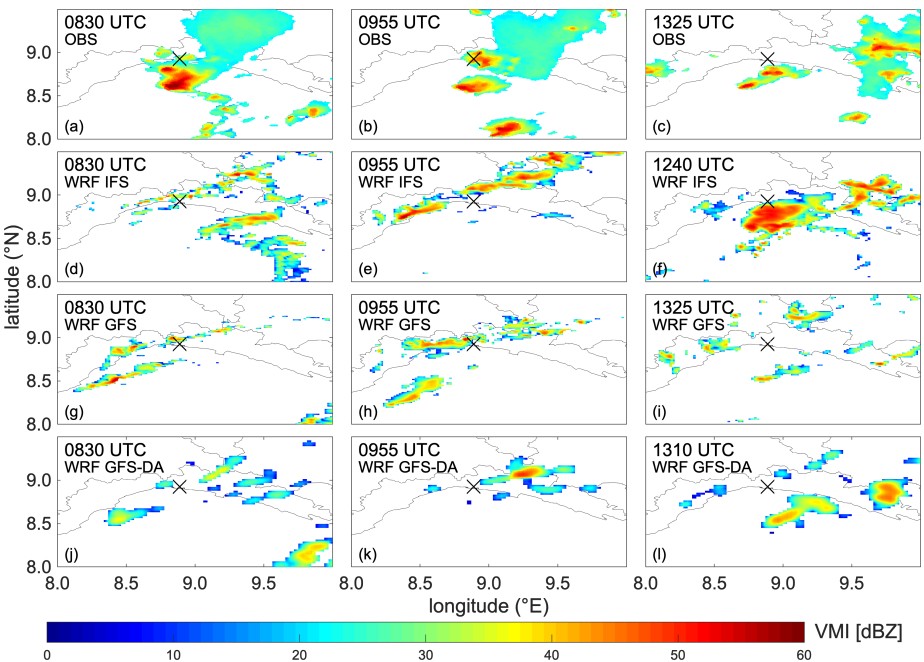

**Figure 17. Radar reflectivity (dBZ) measured by the Ligurian Doppler radar. VMI maps from observations (a-c) and WRF simulations (d-l). The position of Morandi bridge is indicated by the black cross.**


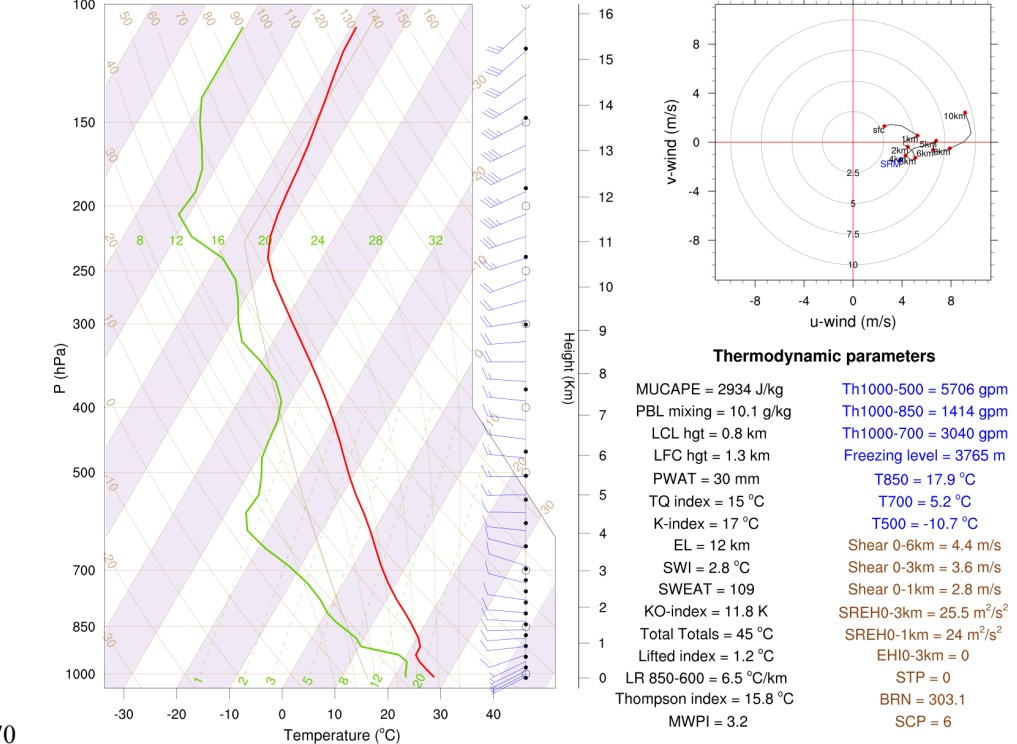

**Figure 18. WRF-IFS thermodynamic diagram at 1240 UTC in the center of the convective structure reported in Fig. 17f.**

