# Peer review of "Investigation of the weather conditions during the collapse of the Morandi Bridge in Genoa on 14 August 2018"

_Natural Hazards and Earth System Sciences, 2019_

## Referee Comment (RC1) · Anonymous Referee #1 · 17 Dec 2019

**Review of "Investigation of the weather conditions during the collapse of the Morandi Bridge in Genoa on 14 August 2018" by Burlando et al. 2019 (nhess-2019-371)**

December 17, 2019

In this study, the authors describe the weather conditions during the period when the Morandi Bridge in Genoa collapsed. They use observational data from satellite, radar, lightning detection sensors, surface stations and a Doppler lidar. Based on the lidar data, the authors characterize the gust front associated with the passing thunderstorm. In addition they performed model simulations with three different forcing conditions and compared the modeled reflectivity and wind speed at the surface with the observed radar reflectivity and surface wind measurements. The authors conclude that a downburst occurred at the time of the bridge collapse a few kilometers from the bridge. However, there is no clear evidence that atmospheric conditions played a role in the bridge collapse.

The motivation for this study was the collapse of the Morandi Bridge. This is without doubt a very tragic event which needs to be investigated to identify the causes for the collapse. So far there is no evidence that the atmospheric conditions around the time of the collapse played a role in the collapse and the presented analysis does not provide any new insights into this. Neither maximum wind speed nor precipitation amounts were extreme. No substantial conclusions are drawn and considering this the scientific impact of the study is rather weak. The study further lacks a clear scientific objective as the description of atmospheric conditions is not an objective. The model simulations also are not analysed in detail and do not add any new insights into the relation between the bridge collapse and weather conditions. I find it very surprising that the simulation with the data assimilation (WRF GFS-DA) totally misses the storm while the simulation WRF-IFS fits much better to the observation. The analysis of the measurements should be more integrated and needs to be improved at several points (further details are given in the specific comments) e.g. the authors should try to link the observations from the different sources in a better way (e.g. wind measurements from the surface stations and the lidar) and provide a more in depth analysis. Because of all these considerations, I cannot recommend the publication of this article in its current state and I believe that the necessary modifications are too extensive to be done within a major revision. However, the study includes some interesting and new parts (like the Doppler

lidar measurements), which I believe deserve publication and I encourage the authors to resubmit a completely revised version of the manuscript with a changed focus. One possible way could be to focus on the gust front observed by the Doppler lidar and use the supplemental measurements from surface stations, radar and satellite to support the hypotheses on the gust front obtained from the Doppler lidar, e.g. propagation speed and direction. The link to the collapse of the bridge could of course be mentioned but should not be the motivation for the study. This could rather be the observational capturing of a gust front by Doppler lidar measurements. In their new manuscript, the authors should try to reduce the number of references to the most relevant (at the moment the number of citations is very large). The number of figures is too large as well and the authors should try to reduce them by combining measurements from different sources. While the English writing is mostly fine, I recommend the manuscript to be checked by a native speaker (e.g. for the correct usage of articles).

**1 Specific comments**

1. Title: The title is misleading as it suggests a link between the weather conditions and the bridge collapse.

2. l. 41: While the authors state that there is lightning in the picture, the lightning was not detected by the network (as described in Sect. 3.1).

3. l. 53-56: This may be the case but the presented data are not really suitable to quantify the role of weather in the collapse.

4. l. 59-62: The large HyMeX programme should be mentioned in this context.

5. l. 62-66: In my opinion it is not necessary to describe the previous storm in so much detail as it is not decisive for the study.

6. l. 76-78: The comparison between downbursts and the ABL is not adequate. The statement implies that the ABL wind is correctly represented in models. Models still fail to correctly represent the ABL conditions in particular over complex terrain. As the scale of downbursts is not resolved by NWP grid spacing, it is not surprising that the models fail to represent them.

7. l. 79-86: Confine to the most relevant studies.

8. l. 101-106: The detailed description of aircraft crashes is not relevant.

9. l. 111: This is not a scientific objective.

10. l. 115ff: The model simulations are not presented in a way that they complement the observational analysis.

11. l. 140: What does LAMPINET stand for?

12. l. 162: What does THUNDERR stand for?

13. l. 221: Where are Querida I and II in Fig. 4a. I don't see Querida over northwestern Scandinavia but rather northeastern.

14. l. 233: I doubt that the connection of the two clusters over the sea is really relevant for the gust front observed at the coast.

15. l. 237: The cloud tops above Genoa already reached 12.000 m at 09:15 UTC according to Fig. 5a.

16. l. 241-248: The movement of the cells is not very visible in Fig. 6. The x- and y-axis range should be reduced. How do the radar images fit to the satellite observations? The shape of the cells is quite different. How can this be explained? What is the vertical structure of the cells? Instead of VMI the authors could show a MaxCappi.

17. l. 248ff: I cannot distinguish the impact of orographic channeling on the precipatation cell. There is no observational evidence for this in the presented figures.

18. l. 258: The SRI images in Fig. 7 are rather redundant to Fig. 6. How are the translation velocities calculated, i.e. how is the arrow drawn in the figure at 09:00 UTC calculated?

19. l. 264: north-eastward!

20. l. 265ff: The evolution of the cell in Fig. 7 does not agree with the statement in l. 263-264 that the cell is more compact over orography.

21. l. 272-274: The lightning distribution does not fit to the reflectivity images in Fig. 7. Please explain.

22. l. 281-282: This is in contradiction to the statement on lightning in the introduction.

23. l. 287ff and Fig. 9: Wind direction should be plotted as markers and not as a connected line as this is misleading. Markers should also be added to temperature and wind speed as the temporal resolution is low. Wind direction might as well change over north. This is not clear from this measurement as the temporal resolution is too low. How does the northerly wind observed at the surface fit to the northward propagation of the cells? Why is temperature not shown for all stations where it is available (Tab. 1). It would be very interesting to see the spatial distribution of the drop in temperature.

24. l. 308: I do not see a subsequent increase in pressure.

25. l. 318: How are hail and graupel distinguished at the weather stations?

26. l. 320ff: Be more specific. Station 11 shows westerly wind and stations 9 and 8 show southerly

wind. For a clearer presentation of the spatial distribution of wind the authors could show a spatial map with the wind plotted as arrows at the individual sites for specific times before, during and after the downburst.

27. l. 352: Note the recent study of Pantillon et al. (2019) on the observational detection of downbursts with Doppler lidar.

28. l. 358: How is the region with maximum velocity identified? Did the authors use an objective method?

29. Fig. 12: This figure needs to be much improved. The shown range has to be adapted to the actual measurement range and color range also needs be decreased. The coast line should be added and maybe even the surface measurements at the caostal stations. The time stamp format is confusing. Is it really necessary to show the seconds?

30. l. 361ff: It is important to realize that radial velocity is measured by the lidar. This means that the three-dimensional wind vector is projected on the direction of the lidar beam. For example, southerly wind cannot be detected at easterly azimuth angles. This means that the front could be wider, but not be detected anymore by the lidar due to a changing azimuth angle. Also, the decrease of radial velocity at larger elevation angles could be related to a changing wind direction. Overall, the problems and difficulties related to radial velocity mesurements need to be much more discussed.

31. Fig. 13: How is it possible that eight gust front heights are given in Fig. 13, while only 7 are indicated in Fig. 12? From which elevation angles is the displacement velocity determined? The contribution of the horizontal wind to the radial velocity is different at different elevation angles. Could this play a role in the estimated gust front heights, i.e. why is the gust front at 7.5 degree always higher than at 10 degree?

32. l. 398-400: From which station are the surface measurements taken? Are the values similar when using the other stations measuring temperature (Tab. 1)?

33. l. 408-409: The values in Charba (1974) were very different compared to the ones found in the present study. It would be worth discussing the different atmospheric conditions under which the gust fronts occurred.

34. l. 420-421: Here the authors state that the wind speed increases with height in the outflow. However, radial velocity decreases with increasing elevation angle, i.e. with height according to the detected gust front heights (Fig. 13). Please explain.

35. l. 449-450: Where can the two regions with high wind speed be found in Fig. 14?

36. l. 451-452: "decreases with time": At which stage of the gust front? Prior or after the the maximum of the gust front was reached? Distances given are along the beam? This needs to be converted to horizontal distance which is different for different elevation angles. According to Fig. 13 the gust front height is largest for 7.5 degree elevation angle (and not 10 degree).

According to this the distance would decrease with height. This whole issue needs to be analyzed more carefully paying more attention to horizontal and along beam distances and height compared to elevation angle.

37. l. 463-464: How does this easterly wind detected by the Doppler lidar relate to the surface measurements? How does it relate to the northward propagation of the gust front. Please discuss!

38. l. 475-476: Which two peaks? There are no peaks visible in WRF-GFS and WRF-GFS-DA in Fig. 16. Why is the period confined to 08:30 and 09:55 UTC? Why not before 08:30 UTC when the wind speed was just as high in WRF-IFS? How does the wind direction in the model compare to the observed ones?

39. l. 487-490: The storm at 13:25 UTC in the observations also fits to the storm in WRF-IFS at that time.

40. l. 491ff and Fig. 18: Why is the thermodynamic diagram calculated for the grid point in the center of the cell? Thermodynamic diagrams are usually used to describe the pre-convective or post-convective environment and not the conditions within the cell. If the grid point shown in Fig. 18 is within the cell, why is there no saturation?

41. l. 506ff: The analysis of the simulations is rather superficial and at least for the WRF-IFS run which is able to produce some storms in the area a comprehensive evaluation of the model output for temperature and wind should be performed and the shape and existence of a potential gust front should be analysed. I cannot follow the conclusions the authors draw from the model simulations, e.g. I do not see a potential of the WRF runs to provide precursors for storms as two of three simulations fail to produce any kind of storm and the third simulation represents it at the wrong time.

**References**

Pantillon, F., Adler, B., Corsmeier, U., Knippertz, P., Wieser, A., and Hansen, A.: Formation of Wind Gusts in an Extratropical Cyclone in the Light of Doppler Lidar Observations and Large-Eddy Simulations, Mon. Wea. Rev., doi:https://doi.org/10.1175/MWR-D-19-0241.1, 2019.

---

## Referee Comment (RC2) · Anonymous Referee #2 · 8 Jan 2020

Review of "Investigation of the weather conditions during the collapse of the Morandi Bridge in Genoa on 14 August 2018" by Massimiliano Burlando, Djordje Romanic, Giorgio Boni, Martina Lagasio and Antonio Parodi.

The paper describes the weather conditions in Genoa on the morning of 14 August 2018, when the collapse of a highway bridge resulted in numerous fatalities. Images captured by a security camera close to the bridge show strong winds and lightning during the collapse, which raises the question of a possible contribution of meteorological factors. Based on a diversity of remote sensing observations, including satellite, radar, lightning sensors and Doppler lidar, as well as a dense local network of surface sta-

tions, the paper highlights the presence of a thunderstorm located along the coast and of winds reaching 15 m/s due to the associated outflow. The event is hardly captured by convection-permitting WRF simulations due to its complex and local dynamics.

The paper describes a convective event based on comprehensive observations from a variety of sources. It presents interesting new material and is well written overall. However, it suffers several major flaws in its present form. The scientific topic of the paper is unclear and the main conclusions are not sufficiently supported. Perhaps related to the first issue, the discussion often spreads in diverging directions instead of focusing on relevant content, which renders the interpretation confusing or even speculative. In particular, the section based on model simulations is not convincing. General and specific comments are listed below to help alleviating these flaws. The paper thus requires substantial revision before it can be considered for publication in Natural Hazards and Earth System Sciences.

GENERAL COMMENTS

1. What is the scientific topic of the paper? A hypothetical contribution of meteorological factors to the bridge collapse would involve engineering considerations and goes far beyond the scope of the paper. Furthermore, the results rather suggest that the wind was unexceptional, which is not clearly stated in the abstract and conclusions. The meteorological questions must be better introduced and the general knowledge that is acquired from that specific case study must be better highlighted.

2. The interpretation of meteorological data, albeit interesting overall, tends to be confusing and sometimes speculative. In my opinion, the results suggest that two different gust fronts reached the western and eastern stations. This interpretation may be erroneous but the description of results and their presentation, e.g. the different space and time coordinates used for the different types of observations, prevents a clear picture of what happened. The analysis must be improved by emphasizing important information on figures, better connecting the different types of data and avoiding over interpretation.

3. The configuration of model simulations does not look appropriate for the event and their contribution to understanding its dynamics is very limited. Either run new model simulations and analyze results in details, or remove the section altogether.

SPECIFIC COMMENTS

Abstract

l. 25-26 Operational predictability cannot be discussed without using operational forecasts.

1. Introduction

l. 28-33 Beginning with a discussion of bridge design is not appropriate in a geoscience journal.

l. 33-35, 117-122 Please clarify; I understand you want to be careful but the message is confusing as it is.

l. 49-56 These claims need references.

l. 57-68 The paragraph does not fit with the topic: thunderstorms are not usually associated with cyclones in the mid latitudes; this may be different in the Mediterranean area but needs justification; the cited Zolt et al. (2006) does not mention downbursts.

l. 78-90 The list of publications does not need to be exhaustive and must be shortened; the discussion of the number of Google Scholar publications is not relevant to motivate the study.

l. 99-105 It would be more logical to introduce these historical papers first, then the more recent examples, and finally the systematic bibliography above.

2. Data and Numerical Simulations

l. 181-191 Why use two very different domains? The first one looks unnecessarily large for a thunderstorm case study.

3. Results and discussion: observations

l. 217-230 The two low pressure systems over northern Europe are not relevant for the study (and the dates do not match); better focus on the region of interest and emphasize contributing factors, e.g., instability, cyclogenesis and fronts.

l. 231-239 I do not clearly see the formation of a convective line; changing the color bar (16 km is reached in the tropics only) and improving the overall poor quality of Figure 5 may help.

l. 243-257 This discussion appears speculative.

l. 277-278 This period includes the collapse time and should thus be shown and discussed.

l. 284-310 Figure 9 needs improvement for interpretation: sampling of "20–30 min" and "approximately 2.5 hours" are confusing and time labels every 2:24 h are weird; better zoom on the time of interest and show every single point of measurement, including a time series of wind gusts rather than a single value; the observed impact on pressure is very speculative and is better omitted; finally, the temperature and wind change occurs over a period of about 1 h, which is not "abrupt" and does not support the presence of a macroburst.

l. 305-308 This belongs to the introduction.

l. 311-314 The difference between coastal and terrain station is more striking than between west and east; drop symbols are illustrative but not quantitative, numbers are also needed.

l. 319-341 The interpretation is not convincing: eastern and western stations clearly behave differently, which is consistent with the location of the convective cell over the eastern stations; however, the slow wind turning at western stations does not support the presence of a macroburst; moreover, the claimed association with a gust front is confusing without a spatial representation; what about temperature records at nearby

stations?

l. 341-346, 372-374 This belongs to the introduction; a definition of downburst, macroburst and gust front is lacking and associated cold pools should be explained as well.

l. 356-371 This discussion is lengthy and should be streamlined; zooming in would help identifying features on Figure 12, which is white mostly (no data).

l. 374-385 (and 422-424) The eight symbols in Figure 13 depict eight different times, thus their representation is confusing and the computation of a displacement velocity obscure (and of the inclination angle).

l. 386-429 Considering the small contribution of moisture, the uncertainty in k and the very speculative increase in pressure, the computation and discussion should be largely simplified.

l. 430-455 This discussion is largely speculative and should be streamlined or omitted.

l. 456-463 What can be learned from Figure 15?

4. Results and discussion: WRF Numerical Simulations

l. 465-517 The purpose of this section is unclear: even the best run (WRF-IFS) still strongly differs from observations thus would need a much more detailed and systematic analysis to provide useful information on the actual dynamics of the event, while the model configuration is too different in the other two runs to provide useful information about model sensitivity with such a local event.

5. Conclusions

l. 551-556 The study clearly shows the presence of a thunderstorm during the storm collapse but rather suggests that associated winds were not extreme; how these may have or not have affected the bridge is far beyond the scope of the study.

---

## Author Comment (AC1) · 8 Jan 2020

**Answer to Reviewer #1 general comment**

We thank the reviewer for thoroughly examining our manuscript and providing many useful comments. This answer addresses only the reviewer's general comment, while the answers to specific comments will be provided separately in the revised version of the paper.

Our study investigates weather conditions during the collapse of the Morandi Bridge in Genoa using various surface and remote sensing measurements, as well as WRF model results. While there is no clear evidence that the weather conditions played a role in the bridge collapse, which we have clearly stated in the manuscript, the reported weather conditions were characterized with strong near-surface winds and high frequency of lightning strikes, in addition to locally strong rainfall rates. These high impact weather conditions could have been a factor in the collapse and therefore it is important to document them in the scientific literature. Now, for some natural phenomena, like landslides or earthquakes, a direct cause-effect relationship between forcing and structural failure can be proven more easily. When it comes to wind, it is more difficult to prove that the weather conditions (e.g., strong winds) triggered the collapse.

A detailed structural analysis would have to be carried out in order to determine precisely if the observed strong winds caused the bridge collapse. In principle, this would include the following analyses: (1) structural integrity of the bridge and (2) the characterization of external forces that were exerted on the bridge—i.e., wind in this particular case. While the first kind of analysis is in this moment ongoing as part of a governmental judicial inquiry, the spatiotemporal characterization of the wind field is one of the main objectives of this article. Our paper extends the analysis of weather conditions beyond wind and includes lightning and rainfall as both of these can be classified as high impact weather. In other words, a study like this one needs to be prepared in order to ever investigate if wind could have caused bridge collapse.

Therefore, in our opinion the objectives of the paper are clearly defined and scientifically sound. It is not that bridges are falling down on daily basis around the world and then this is just one more paper on that subject. This tragic event is a rare incident whose occurrence coincides with the development of a strong thunderstorm in the same area. That being said, we do not see how the investigation of that thunderstorm is not important from both meteorological (phenomenological) and, in perspective, engineering (structural) point of view.

In addition, this manuscript is submitted to the journal of *Natural Hazards and Earth System Sciences* (NHESS) and not to purely structural engineering or meteorological journals. The aims and scope of NHESS (https://www.natural-hazards-and-earth-system-sciences.net/about/aims_and_scope.html) are:

- *the study of the evolution of natural systems towards extreme conditions, and the detection and monitoring of precursors of the evolution;*
- *the detection, monitoring, and modelling of natural phenomena, and the integration of measurements and models for the understanding and forecasting of the behaviour and*

*the spatial and temporal evolution of hazardous natural events as well as their consequences;*

- *the design, development, experimentation, and validation of new techniques, methods, and tools for the detection, mapping, monitoring, and modelling of natural hazards and their human, environmental, and societal consequences;*
- *the design, implementation, and critical evaluation of mitigation and adaptation strategies to reduce the impact of hazardous natural events on human-made structures and infrastructure, to reduce vulnerability and to increase resilience of individuals and societies;*
- *the analysis of the impact of climatic and environmental changes on natural hazards and their consequences."*

While most of the above-listed subjects can be found in our manuscript, the second bullet point (underlined) is the most relevant for our research. Our paper presents detection, monitoring and modelling of natural phenomena (i.e., thunderstorm) through the integration of measurements (surface and remote measurements) and models (WRF) to properly understand the forecasting potential and behavior of the spatial and temporal evolution of this potentially hazardous natural event. Most of the general comments made by the reviewer suggest a purely meteorological analysis with little to no mentioning of the bridge collapse. Such an analysis, however, would not fit this journal and it would be more appropriate for meteorological journals.

While we agree with some of the reviewer's comments, we believe that the provided review is not adequately formulated keeping in mind the scope of this journal, the scope of this research and the overall objectives that the authors set for the manuscript. For example, the reviewer states: "*The link to the collapse of the bridge could of course be mentioned but should not be the motivation for the study. This could rather be the observational capturing of a gust front by Doppler lidar measurements.*" We have two main objections to this statement. Firstly, the reviewer is stating what the objective of our own research should be. We can accept that the reviewer is not satisfied with the methods, research level, presentation of results, quality of figures, etc., but the review, in principle, should not reflect on how the reviewer would formulate the study's objectives if she/he carried out this research. We can all agree that different people have different research interests and our interests in the current stage of this research project are to describe, quantify and understand the genesis and predictability of this severe weather event that took place during the bridge collapse—all within the scope of this journal.

It is true that at the end of this investigation we cannot state that for sure wind was the cause of bridge collapse, also because there are no wind measurements at the bridge site to demonstrate directly this hypothesis. However, we have shown that exactly at the time of collapse a gust front was surely passing over the bridge, as demonstrated qualitatively by Figure 1 and quantitatively by the analysis reported in Section 3.2. We think that assuming this is just a coincidence would be too superficial and therefore we cannot agree with the Reviewer's statements that "there is

no evidence that the atmospheric conditions around the time of the collapse played a role in the collapse" and "the presented analysis does not provide any new insights into this".

The Reviewer states also that "Neither maximum wind speed nor precipitation amounts were extreme". However, natural hazards are not only related to very strong or extreme events. In a broader sense of its meaning, a natural hazard can be any cause of disaster, which is not always an extreme event. In other words, many structures are nowadays old and therefore susceptible of collapse because of fatigue. Under such assumption, even not really extreme events, like the one described in our paper, can become a natural hazard when it affects a structure which is very old and not well maintained or monitored, as it was the case of Morandi Bridge.

Lastly, we would like to address the general comment on the number of figures and English. Most articles today contain between 10 and 20 figures, and we have included 18 figures in our paper. We believe that all figures are relevant because each of them shows different aspects of the research results. We also find confusing that the reviewer suggests here that the number of figures is too large, but then additional figures are recommended later in the specific comments (e.g., temperature for all stations and spatial map of wind vectors). We agree with the reviewer that there are some English language inaccuracies and these will be corrected in the revised manuscript. Thank you.

---

## Author Comment (AC2) · 1 Mar 2020

**Answers to Reviewer #1**

**Specific comments**

**Q1:** Title: The title is misleading as it suggests a link between the weather conditions and the bridge collapse.

**A1:** We kindly disagree with the reviewer's observation that the title suggests there is an established connection between the collapse and the weather conditions. It is a fact that these weather conditions were observed during the bridge collapse. Hence, we wrote "…weather conditions during the collapse…" and not weather conditions that caused or potentially caused the collapse. Therefore, we have left the title as it was in the original manuscript.

**Q2:** 41: While the authors state that there is lightning in the picture, the lightning was not detected by the network (as described in Sect. 3.1).

**A2:** The lightning was detected by the network indeed. Fig. 1a is taken from the west with respect to the bridge and the photo shows the east. This is indicated in Fig. 1b because if north is left and south is right, east is forward. Now, in Fig. 8 you can clearly see that lightning strikes were detected to the east of the bridge; therefore, everything is perfectly coherent.

**Q3:** 53-56: This may be the case but the presented data are not really suitable to quantify the role of weather in the collapse.

**A3**: We struggle to understand the reviewer's comments on why it is not good that we did not explicitly demonstrate that the weather caused the collapse. As we explained in our answer to the general comment, in order to quantify the wind actions on the bridge, the person would have to carry out a detailed structural analysis study. However, such a study is impossible to be conducted if there is no information about the wind. Hence, this study documents and explores the wind conditions in the thunderstorm that was present during the collapse.

**Q4:** 59-62: The large HyMeX programme should be mentioned in this context.

**A4:** The paper by Drobinski et al. (2013) will be included in the citation list. Thank you.

**Q5:** 62-66: In my opinion it is not necessary to describe the previous storm in so much detail as it is not decisive for the study.

**A5**: This description will be shortened to a single sentence in the revised manuscript.

**Q6:** 76-78: The comparison between downbursts and the ABL is not adequate. The statement implies that the ABL wind is correctly represented in models. Models still fail to correctly represent the ABL conditions in particular over complex terrain. As the scale of downbursts is not resolved by NWP grid spacing, it is not surprising that the models fail to represent them.

**A6:** This sentence does not mention numerical modelling—the NWP or other modelling—of either ABL or downburst winds. We simply state that the dynamics of downbursts are more complex that the dynamics of ABL winds.

**Q7:** 79-86: Confine to the most relevant studies.

**A7:** This section will be significantly reduced. Only a sample of few relevant studies for field measurements of downburst winds in the Ligurian and the northern Tyrrhenian Sea will be included. The references to Google Scholar indexing of downburst-related studies will be also omitted in the new manuscript.

**Q8:** 101-106: The detailed description of aircraft crashes is not relevant.

**A8**: This part will be removed from the revised manuscript.

**Q9:** 111: This is not a scientific objective.

**A9**: We kindly disagree with the reviewer. As we explained in our answer to the general comment above, this objective corroborates with the scope of this journal. The paper investigates a severe natural phenomenon that occurred during a hazard. It is certainly not illogical to suspect a connection between high impact weather (thunderstorm downburst) and bridge collapse, without attempting to dis(prove) such a connection. This bridge collapse is such an enormous event that will probably be investigated in many studies and it is impossible to lump together all different research approaches in one paper. While we are planning to analyze this event from different perspectives too (some of them being similar to what the reviewer suggested here), these studies are beyond this paper. The present paper rigorously describes the weather conditions during the collapse, as well as the contributing factors for this severe weather.

However, we will add the sentence that the future study that we are currently working on will present a detailed three-dimensional reconstruction of this flow field using Doppler lidar and radar observations. This study is still ongoing and the scope of such a paper is probably not adequate for this journal.

**Q10:** 115ff: The model simulations are not presented in a way that they complement the observational analysis.

**A10**: The numerical simulations were conducted in order to investigate the predictability of this thunderstorm event in an operational weather forecasting model. We opted not to use any research-oriented WRF configurations, but instead, we wanted to assess the accuracy of the operational weather forecast of this weather event in the given 1-h interval around the bridge collapse.

In addition to the original manuscript version, we have now carried out a simulation including the data assimilation on the IFS driven forecast, namely WRF-IFS-DA. This will be added to the discussion in the revised paper. This prediction in comparison to the WRF-IFS experiment shows two main advantages: (1) wind peaks up to 12–13 m s$^{-1}$ over the 8 wind stations used in the study; and (2) the thunderstorm event occurring one hour closer to the observed wind peaks. The forecast improvements are due to a better modelling of the observed thunderstorm activity (Figure A) close to Genoa in the late morning of 14 August 2018. Therefore, the WRF-IFS-DA simulation gave better results than the other runs. The WRF-IFS-DA predicted the storms in the period 11–12 UTC and produced the maximum wind gusts around 15–20 m s$^{-1}$ (Figure B). The modelled storm was located only 8–10 km away from the location of anemometers and the location error is only 4–6 times the grid spacing (Grasso 2000). Indeed, this is a very accurate simulation result.

[Figure]

Figure A. VMI map from WRF-IFS-DA run at 11:30 UTC.

[Figure]

Figure B. Maximum 5-min wind speeds at 10 m above ground from the WRF-IFS-DA simulation at the locations of eight anemometers in the Port of Genoa.

All considered, it is now even more relevant the role of the numerical results to complement the observational analysis, in particular after the additional effort to make the WRF-IFS-DA products. We

believe that demonstrating that only single members of the operational WRF suite properly capture this event in terms of its location, but not necessarily timing, is as scientifically significant as if all WRF configurations provided a good comparison against the observations. Furthermore, the new simulation is sufficiently capturing this thunderstorm event and therefore enables a deeper analysis of the physical processes inside and outside of the cloud. This discussion will be added to the revised manuscript.

**Q11:** 140: What does LAMPINET stand for?

**A11:** LAMPINET stands for the Lightning Network of the Italian Air Force Meteorological Service. This will be included in the revised manuscript.

**Q12:** 162: What does THUNDERR stand for?

**A12:** THUNDERR is an acronym of the project Detection, Simulation, Modelling and Loading of thunderstorm Outflows to design wind-safer and cost-efficient structures. THUNDERR is an abbreviation of THUNDERstorm that expresses the innovative Roar of this research. This description will be included in the footnote in the revised manuscript.

**Q13:** 221: Where are Querida I and II in Fig. 4a. I don't see Querida over northwestern Scandinavia but rather northeastern.

**A13:** This was a mistake, Querida is over northeastern Scandinavia indeed. Thank you.

**Q14:** 233: I doubt that the connection of the two clusters over the sea is really relevant for the gust front observed at the coast.

**A14:** This analysis is looking at this thunderstorm system from a different (larger) scale. The goal here is not just to characterize the gust front and smaller-scale thunderstorm phenomena, but also to investigate the larger-scale weather scenarios that were associated with the thunderstorm development and propagation between 09:00 UTC and 10:00 UTC.

The connection between the thunderstorm development at the large scale and the gust front was not made in the manuscript, whereas this comment indicates that we have linked these two. So, we would like to clarify that too. Yet another goal of this analysis was also to show that this cloud system was slowly evolving and the cloud cover above the Genoa region stayed effectively unchanged throughout the collapse hour.

**Q15:** 237: The cloud tops above Genoa already reached 12.000 m at 09:15 UTC according to Fig. 5a.

**A15:** We thank the reviewer for this comment because we poorly expressed ourselves here. We did not want to say that some of the cloud tops were not at these heights prior to 09:45 UTC, but we wanted to say that the clouds were at 12,000 m at 09:45 UTC regardless of their previous top height. We will revise the sentence for clarity in the new manuscript. Kindly note that Figure 5c (below) now will also contain a zoom-in of the cloud tops above Genoa (also see our answer A16).

[Figure]

**Q16:** 241-248: The movement of the cells is not very visible in Fig. 6. The x- and y-axis range should be reduced. How do the radar images to the satellite observations? The shape of the cells is quite different. How can this be explained? What is the vertical structure of the cells? Instead of VMI the authors could show a MaxCappi.

**A16:** The range of *x* and *y*-axis will be reduced in Figure 6.

The radar data in Figure 6 are shown over a much smaller region than the satellite observations in Figure 5. However, both figures show that there is a small region of very shallow clouds south of Genoa. This area is now indicated with a red arrow in the new Figure 5c (see figure above in A15). This discontinuity is

also observed in the radar measurements in Figure 6. In addition, the pattern of radar reflectivity does not necessarily have to match the spatial pattern of could tops. An example is the Mesoscale Convective Complexes, in which case the cloud tops have a circular shape, but the radar reflectivity can resemble a squall line or different cloud cluster (Markowski and Richardson, 2010).

We do not have the vertical volume slices for this event in order to describe the vertical structure of cloud cells. The MAXCAPPI and few other radar products were excluded in order to have more space for other analyses in this manuscript. The reviewer also suggested that the number of figures could be reduced. Including a MAXCAPPI figure would only add more material and make the manuscript larger. Lastly, we believe that the presented radar products in terms of reflectivity (VMI) and surface rain intensities are sufficient for the analyses presented in this paper.

**Q17:** 248ff: I cannot distinguish the impact of orographic channeling on the precipitation cell. There is no observational evidence for this in the presented figures.

**A17**: The northeastern part of the VMI reflectivity in Figure 6 (VMI around 30 dBZ) follows the direction of the orographic valley. On the other hand, the orientation of the two cells that are located above the sea is zonal. The curvature in the VMI footprint in Figure 6 coincides with the location of the valley and it led us to propose that orography had an influence on cloud development. Of course, the orography, river valleys, and shorelines are the well-known contributors for cloud development and propagation.

We will implement this comment in the revised manuscript by stating that it is likely the orography influenced the cloud propagation.

**Q18:** 258: The SRI images in Fig. 7 are rather redundant to Fig. 6. How are the translation velocities calculated, i.e. how is the arrow drawn in the figure at 09:00 UTC calculated?

**A18**: This figure shows the rain intensity while Figure 6 is the radar reflectivity. We argue that Figure 7 is complementary to Figure 6 and not redundant.

The arrow in Figure 7 uses the direction and magnitude of the speed of the rain cell centroid derived in the following way.

The 5-min Surface Rainfall Intensity (SRI) estimates were obtained from the national mosaic of the Italian Meteo Radar Network (see http://www.protezionecivile.gov.it/risk-activities/meteo-hydro/activities/prediction-prevention/central-functional-centre-meteo-hydrogeological/monitoring-surveillance/radar-map for examples and additional details) and were used to identify the storm cell movement towards the coast in the period from 9:00 to 9:55 UTC on 14 August 2018.

The cell was isolated from the rest of the precipitation field observed at the national scale through the application of a region growth algorithm to the binary image rain/no rain (Gonzalez and Woods, 2002). For each time step, $t_i$, the centre of mass of the precipitating cell was identified and the module, $v_{t_i}$, and direction, $\theta_{t_i}$, of the cell were calculated by:

$$v_{t_i} = \frac{\left| \vec{r}_{t_{i+1}} - \vec{r}_{t_i} \right|}{\Delta t}$$

$$\theta_{t_i} = \tan^{-1} \left( \frac{x_{t_{i+1}} - x_{t_i}}{y_{i+1} - y_{t_i}} \right).$$

Here, $\vec{r}_{t_i} = (x_i, y_i)$ is the position vector of the cell centroid with the longitude, $x$, and the latitude, $y$, in meters.

**Q19:** 264: north-eastward!

**A19:** Thank you for spotting this inaccuracy. It will be corrected in the revised manuscript.

**Q20:** 265ff: The evolution of the cell in Fig. 7 does not agree with the statement in l. 263-264 that the cell is more compact over orography.

**A20**: The cell is not more compact over land, but we observe that it gets elongated in the direction of the valley after it makes its landfalls at Genoa. A similar pattern of elongation is also observed in the reflectivity data in Figure 6. We do agree that Figure 6 and 7 do not look identical, but it is expected because they are not portraying the same quantity. This is an additional reason for keeping both figures in the manuscript.

**Q21:** 272-274: The lightning distribution does not fit to the reflectivity images in Fig. 7. Please explain.

**A21**: It is not clear to us what the reviewer refers to with the following statement: "The lightning distribution does not fit the reflectivity..." The Blitzonturg imagery of lightning seems to follow nicely the meridional orientation of cloud tops that were also captured in the satellite images. On the other hand, the lightning captured by the LAMPINET network is mostly located south to southeast from the bridge (northeastward motion of the lightning zone). Note that the LAMPINET imagery is only a subset of the Blitzonturg data that covers a much larger region. The northeastward propagation of the lightning zone is similar to the overall propagation of the precipitation zone in Figure 7—i.e., both are predominantly northeastward.

**Q22:** 281-282: This is in contradiction to the stdatement on lightning in the introduction.

**A22**: We have clarified this part because we believe it is very important for the paper. In Line 41 it is stated that "the lightning occurred to the east-southeast of the bridge", not over the bridge, and this is coherent with the statement in lines 281-282, as also already explained in the previous answer A2.

**Q23:** 287ff and Fig. 9: Wind direction should be plotted as markers and not as a connected line as this is misleading. Markers should also be added to temperature and wind speed as the temporal resolution is low. Wind direction might as well change over north. This is not clear from this measurement as the temporal resolution is too low. How does the northerly wind observed at the surface fit to the northward propagation of the cells? Why is temperature not shown for all stations where it is available (Tab. 1). It would be very interesting to see the spatial distribution of the drop in temperature.

**A23**: Wind direction is now plotted as markers. Markers are also added to temperature and wind speed time series. The markers now show that the wind direction was indeed changing in the counterclockwise direction in the interval centered on the collapse time (between 08:50 UTC and 10:20 UTC). This will be clarified in the manuscript text too.

The wind direction at and around the bridge collapse time was not from the north. The wind direction was 140° (southeast). In relation to the bridge collapse hour, the northern winds are only observed around 08:50 UTC and around 10:30 UTC.

Air temperature is not shown for all stations in order to shorten the manuscript and reduce the total number of figures. Moreover, also the other temperature measurements are unfortunately available every 30 min and such a low temporal resolution does not allow to add any relevant information concerning the gust propagation. These changes are shown in the revised figure below.

[Figure]

**Q24:** 308: I do not see a subsequent increase in pressure.

**A24**: The maximum value of surface pressure was observed close to the bridge collapse time (Fig. 9c). Unfortunately, the time resolution of pressure data is even lower than temperature and wind data, but nevertheless, we do observe the peak in surface pressure at 09:00 UTC. In addition, it is logical to assume that the pressure did not plateau after 09:00 UTC and immediately started dropping to the next observed value at 12:00 UTC.

Moreover, the gust front theory proposed by Wakimoto (1982) and later nicely depicted in Markowski and Richardson (2010) demonstrates that the pressure rise associated with the thunderstorm (i.e., gust front) passage is slower than the abrupt changes in air temperature or wind velocity. For instance, see Figure 5.23 from Markowski and Richardson (2010).

**Q25:** 318: How are hail and graupel distinguished at the weather stations?

**A25**: There was no need to distinguish between hail and graupel because no ice particles were observed at the surface. We have rephrased this sentence to read as follows: "No hail or any other ice particles were observed in this period at the weather stations in the Genoa region."

**Q26:** 320ff: Be more specific. Station 11 shows westerly wind and stations 9 and 8 show southerly wind. For a clearer presentation of the spatial distribution of wind the authors could show a spatial map with the wind plotted as arrows at the individual sites for specific times before, during and after the downburst.

**A26**: New pictures/panels will be added to Fig. 11 (not to increase the overall number of figures) with the wind plotted as arrows. We didn't do it before because we thought the explanation of Fig. 11 was clear enough, but probably this was not the case, so it is worth adding new pictures as suggested by the Reviewer.

**Q27:** 352: Note the recent study of Pantillon et al. (2019) on the observational detection of downbursts with Doppler lidar.

**A27**: This paper will be cited in the revised manuscript. We thank the reviewer for suggesting the relevant literature.

**Q28:** 358: How is the region with maximum velocity identified? Did the authors use an objective method?

**A28**: The region with the maximum velocity is identified by inspecting raw lidar measurements. We did not use any particular mathematical tool in this process because it is obvious from figures and data (numbers) where the high-velocity zones are.

**Q29:** Fig. 12: This figure needs to be much improved. The shown range has to be adapted to the actual measurement range and color range also needs be decreased. The coast line should be added and maybe even the surface measurements at the caostal stations. The time stamp format is confusing. Is it really necessary to show the seconds?

**A29**: Figure 12 is now improved. The color bar range now represents the real measurements and it is spanning from $-7$ m s$^{-1}$ to $+17$ m s$^{-1}$. The radial extent is also reduced in order to show better resolution of valid lidar measurements. The radial distance is now limited to 5492 m away from the lidar because the data beyond that radius are either unreliable or non-existent due to precipitation. In addition, this figure is now saved at a much higher resolution (600 dpi). The modifications are shown in the figure below, but kindly note that the figure is made to be inserted on a landscape page (not a portrait page as in this document).

[Figure]

The color scheme employed in this figure was specifically designed for lidar measurements. The cool colors are associated with negative Doppler velocities (i.e., away from lidar), whereas the warm colors are showing positive Doppler velocities (i.e., towards lidar). The white color shows 0 m s$^{-1}$ Doppler velocities. We believe that this color scheme is rather intuitive and helps in fast identifications of positive and negative velocity zones. The surface measurements at the coastal stations are already shown in Fig. 11.

When it comes to the coastline, Fig. 2 shows that the lidar is located at the tip of the coastline and effectively over the sea. The azimuth range is from 101° to 250° and it never crosses the coastline. In order to facilitate this comment, we will add the lidar scanning area to Figure 2.

The flow field evolution and the scanning pattern of the lidar are on the time scales of a minute or so. Therefore, we decided to leave the exact times—including seconds—as the lidar software outputs them. It just increases the precision of the reported data. Instead of HH:MM, we present HH:MM:SS, so we do not see the addition of seconds makes these plots more confusing. Since the entire range of observations in Fig. 12 is taking place at 09:MM:SS, we have decided to remove 09 from the label.

**Q30:** 361ff: It is important to realize that radial velocity is measured by the lidar. This means that the three-dimensional wind vector is projected on the direction of the lidar beam. For example, southerly wind cannot be detected at easterly azimuth angles. This means that the front could be wider, but not be detected anymore by the lidar due to a changing azimuth angle. Also, the decrease of radial velocity at larger elevation angles could be related to a changing wind direction. Overall, the problems and difficulties related to radial velocity measurements need to be much more discussed.

A30: The authors are aware of the limitations of lidar measurements and we agree with the reviewer's observations regarding the gust front width. We will correct this statement in the revised manuscript. Indeed, the change of radial velocity could partially be due to the wind direction change with height. However, the changes in the heights between two elevation angles are below 100 m and it is not likely that the wind direction changes drastically in such a shallow layer. In addition, it is not clear from this

comment to which statement on the radial velocity decrease in our manuscript the reviewer is referring to. We have stated that the zone of the high wind speeds is closer to the lidar at the higher elevations.

**Q31:** Fig. 13: How is it possible that eight gust front heights are given in Fig. 13, while only 7 are indicated in Fig. 12? From which elevation angles is the displacement velocity determined? The contribution of the horizontal wind to the radial velocity is different at different elevation angles. Could this play a role in the estimated gust front heights, i.e. why is the gust front at 7.5 degree always higher than at 10 degree?

**A31**: Only seven gust fronts are indicated in Fig. 12 because the front is above lidar around the time instant that corresponds to Fig. 12h (09:21:04–09:21:53 UTC). Even in Fig. 12g the dotted line is very short.

We agree that the contribution of horizontal wind speed to the radial velocity is different at different elevations. The methodology for obtaining Fig. 13 is as follows. Firstly, we determine the distance between the lidar and the gust front edge ($D_f$) at different elevations by inspecting the raw lidar data (Doppler velocities). Then, we project that distance that is along the lidar beam to the horizontal plane ($X_f$) as:

$$X_f = D_f \cos \varphi,$$

where $\varphi$ is the elevation angle. Similarly, the height of the gusts front edge above the lidar ($H_f$) is determined as:

$$H_f = D_f \sin \varphi.$$

Note that the values of $X_f$ and $H_f$ are available for two consecutive scans of the four elevation angles (Fig. 12a–h). We call them $X_{fs1}$ and $X_{fs2}$, and $H_{fs1}$ and $H_{fs2}$ for scans 1 and 2, respectively.

Then, the horizontal displacement velocity of the gust front ($V_{disp}$) at two different heights is calculated as:

$$V_{disp} = \begin{cases} \dfrac{X_{fs1}^{\varphi_1} - X_{fs2}^{\varphi_3}}{6T_b}, & \text{at 115 m,} \\ \dfrac{X_{fs2}^{\varphi_1} - X_{fs2}^{\varphi_4}}{3T_b}, & \text{at 70 m.} \end{cases}$$

Here, the superscript indicates the height at which the horizontal distance is considered and $T_b = 51$ s is the scanning time of lidar (49 s of scanning per elevation plus 2 s of moving to the next elevation angle). These quantities are plotted in Fig 13. Our results show that these two displacement velocities are similar (Fig. 13 and also discussed in the revised manuscript). As demonstrated above, $V_{disp}$ is estimated as the function of geometric variables and not radial (Doppler) velocity along the lidar beam.

**Q32:** 398-400: From which station are the surface measurements taken? Are the values similar when using the other stations measuring temperature (Tab. 1)?

**A32**: The surface measurements are taken from the Genova Airport (i.e., Genova Sestri) weather station. Some of the stations in Table 1 do not record temperature, pressure, and relative humidity (e.g., GEPOA, GEPWA, etc.). Therefore, these stations cannot be used in this analysis.

We have chosen Genova Airport station because it is a registered World Meteorological Organization (WMO) weather station located at the Genoa airport (USAF number 161200) and, in addition, it is the

weather station that is the closest to the coastline. Similar to the lidar, it is effectively measuring the atmospheric conditions above the sea in the cases when the wind direction is from the sea towards the land (as it was in the analyzed case).

The results are not the same if other weather stations are used, but this is expected as the other stations are at higher elevations in comparison to the lidar and Genova Airport sites. Nevertheless, most of the other weather stations in Genoa recorded the minimum temperature around 19°C, but the temperature prior to the thunderstorm gust front passage was not necessarily 23°C, but lower. This indicates that the passage of the gust front was similarly observed at all stations. The air temperature prior to the gust front was lower at other stations because these stations are located at higher altitudes in comparison to Genoa Airport. Indeed, this is one of the major reasons for using the Genoa Airport station (i.e., the same altitude as the lidar, which is effectively the sea level). That is, the elevation of Genoa Airport is 3 m above sea level, whereas the lidar is at 5 m above sea level. The other stations in Table 1 are at much higher elevations. We will add these clarifications to the revised manuscript.

**Q33:** 408-409: The values in Charba (1974) were very different compared to the ones found in the present study. It would be worth discussing the different atmospheric conditions under which the gust fronts occurred.

A33: Indeed, the values in Charba (1974) significantly differ from our values and this was one of the reasons for reporting it in this paper. We would like to emphasize that the methodology presented above (e.g., related to the reviewer's comments Q31 and Q32, and similar) is not proposed and developed to provide the exact values of gust front displacement velocity or vertical structure, but an approximation of these values based on the limited data that are available. Excluding the lidar data, the other data used in this methodology are readily available from standard meteorological measurements.

Charba (1974) analyzed a gust front that occurred on 31 May–1 June 1969 over Oklahoma and Kansas, United States (US). He reported the air temperature drop of about 5°C (25°C to 20°C) and a pressure increase of 0.2 inches of Hg, which corresponds to approximately 677.3 Pa (~6.8 mb). We observe that both temperature and pressure drop in their case was higher than in our study—in particular the pressure drop differences. However, the temporal resolution of our pressure measurements is lower than the resolution of temperature measurements, so the pressure jump in our case could have been higher than the reported value. This is an uncertainty that we clearly reported in Fig. 9. We will add some additional discussion on our results and their similarity to Charba (1974).

However, we kindly emphasize here that Mueller and Carbone (1987) observed the similar value of gust front displacement velocity (i.e., 6.9 m s$^{-1}$) in their study on the dynamics of thunderstorm outflows in the US. We have reported this similarity between the two studies in the original manuscript. Similarly, the manuscript reports the similarity between our results and Goff (1976), as well as Mueller and Carbone (1987).

**Q34:** 420-421: Here the authors state that the wind speed increases with height in the outflow. However, radial velocity decreases with increasing elevation angle, i.e. with height according to the detected gust front heights (Fig. 13). Please explain.

**A34**: Our answer A31 above described the methodology for obtaining the approximate heights of the gust front and its displacement velocity in the direction of the lidar. We have also emphasized now that the

obtain displacement velocity of the gust front is the projection of the overall displacement velocity in the direction of the lidar. Radial velocity is not always decreasing with height (Fig. 12). Quite the opposite, the radial velocity is usually increasing with height above a fixed point. As an example, we fix the point at the radius of 1598 m from the lidar and azimuth of 151°, and we analyze the four different elevation angles in Fig. 12a–d. The radial velocity is clearly increasing with the height (i.e., with increasing elevations). While there are some regions in the flow where the radial velocity is decreasing with height, these are fewer in comparison to the opposite.

**Q35:** 449-450: Where can the two regions with high wind speed be found in Fig. 14?

**A35**: The first region of high wind speed is associated with the leading gust front head, while the second region is located behind the dotted line that indicates the second surge of the cold air that outflows from the thunderstorm. This schematically representation is also in accordance with the gust front outflow model proposed by Charba (1974) in Fig. 11 in his paper. That is, the cold outflows are not necessarily continuously spreading below and in front of the thunderstorm, but rather in a series of surges. We will discuss further this process in the revised manuscript, but note this was also included in the original version of this manuscript. In addition, we have added an arrow and explanation of the second surge in Fig. 13.

[Figure]

**Q36:** 451-452: "decreases with time": At which stage of the gust front? Prior or after the the maximum of the gust front was reached? Distances given are along the beam? This needs to be converted to horizontal distance which is different for different elevation angles. According to Fig. 13 the gust front height is largest for 7.5 degree elevation angle (and not 10 degree). According to this the distance would decrease with height. This whole issue needs to be analyzed more carefully paying more attention to horizontal and along beam distances and height compared to elevation angle.

A36: This sentence is now removed from the revised manuscript. The observation was based on Fig. 12, which shows the results along the beam. The distances between two high-regions in Fig. 12a–d seems to be a bit larger than in the corresponding panels in Fig. 12e–h. However, the closer analysis shows that the decrease is either very small or in some cases not obvious. In order to remove any confusion related to this point, we have decided to remove this part from the revised manuscript.

**Q37:** 463-464: How does this easterly wind detected by the Doppler lidar relate to the surface measurements? How does it relate to the northward propagation of the gust front. Please discuss!

**A37**: This result is in accordance with the cell movement in Fig. 7. The precipitation region was advancing in the northeastward direction and after 09:40 UTC the precipitation zone was located east of the lidar.

Consequently, the outflow that radially spreads from the precipitation zone would approach the lidar from approximately east (i.e., easterly winds). We will clarify this observation in the revised manuscript. Also, kindly note that this figure is now improved in accordance with the reviewer's comment Q29.

[Figure]

**Q38:** 475-476: Which two peaks? There are no peaks visible in WRF-GFS and WRF-GFS-DA in Fig. 16. Why is the period confined to 08:30 and 09:55 UTC? Why not before 08:30 UTC when the wind speed was just as high in WRF-IFS? How does the wind direction in the model compare to the observed ones?

**A38**: We have now added the same plot for the WRF-IFS-DA (Figure B in A10). This experiment shows a clear wind gust peak of 12–13 m s$^{-1}$ at 11–12 UTC at the anemometers' locations. Moreover, the predicted wind directions in WRF-IFS-DA simulation are in good agreement with the observations.

**Q39:** 487-490: The storm at 13:25 UTC in the observations also fits to the storm in WRF-IFS at that time.

**A39**: On 14 August 2018, in the morning there were many convective cells that developed off the coast of Genoa. When looking at all the available time-frames of radar observations (every 10 minutes, not shown in the paper), it is apparent that the two cells in Fig. 17b are the evolution of the larger convective structure shown in Fig. 17a, whereas the two cells shown in Fig. 17c are new cells independent from the previous ones. That being said, we compared the convective structure simulated by WRF-IFS at 12:40 UTC to the one observed at 8:30 UTC just because they are similar in terms of shape, size and position, while the match between WRF-IFS at 12:40 UTC and observations at 13:25 UTC is weaker, in our opinion. However, this does not mean that WRF-IFS at 12:40 UTC and observations at 8:30 UTC are the same convective phenomena and the Reviewer is right saying that the simulated one is also similar, to some extent, to the storm at 13:25 UTC. This will be further clarified in the revised paper.

**Q41:** 491ff and Fig. 18: Why is the thermodynamic diagram calculated for the grid point in the center of the cell? Thermodynamic diagrams are usually used to describe the pre-convective or post-convective environment and not the conditions within the cell. If the grid point shown in Fig. 18 is within the cell, why is there no saturation?

**A40**: The thermodynamic diagram was not calculated for the grid point in the center of the cell. Conversely, it was computed in the region affected by the convective cells transition. However, the diagram is now substituted by the one computed from the WRF-IFS-DA run.

**Q42:** 506ff: The analysis of the simulations is rather superficial and at least for the WRF-IFS run which is able to produce some storms in the area a comprehensive evaluation of the model output for temperature and wind should be performed and the shape and existence of a potential gust front should be analysed. I cannot follow the conclusions the authors draw from the model simulations, e.g. I do not see a potential of the WRF runs to provide precursors for storms as two of three simulations fail to produce any kind of storm and the third simulation represents it at the wrong time.

**A42**: The new simulation is added to complete the set of these figures. The WRF-IFS-DA results show a gust front propagation (see Figure C below) in the period between 11:00 and 11:55 UTC (every 5 minutes). We agree with the reviewer that the model simulations are not able to perfectly reproduce the timing and location of the observed event. However, these simulations are performed in an operational framework and the location and timing error is in line with the state-of-the-art numerical weather modelling prediction of these types of convective events. Furthermore, even with somewhat different timing and location of the event, the WRF-IFS-DA simulation is reproducing the thunderstorm cloud well enough to be used to get a deeper insight into the physics of this event (e.g., analyzing processes that are not obtainable from a surface or remote sensing measurements such as microphysics processes and species).

[Figure]

Figure C: Maximum 10-m wind speed (wind gust) at 5-min temporal resolution in the time period 11:00–11:55 UTC from the WRF-IFS-DA simulation.

As an example, Figure D shows a vertical cross-section of the reflectivity field at 11:30 UTC for the WRF-IFS-DA experiment. Severe convection is apparent in front of Liguria coastlines in association with a wellorganized system of updraft and downdrafts. This system produces near-surface wind gusts that propagate from the south-east (Figure E).

A comprehensive description of this new simulation will be reported in the revised manuscript.

[Figure]

Figure D: Vertical cross-section of the WRF-IFS-DA simulated radar reflectivity along the convective cell developing in front of Liguria coastlines at 11:30 UTC. The shaded colors in the horizontal plan show the air temperature at 2 m above ground.

[Figure]

Figure E: Updrafts (+1 m s$^{-1}$ red isoline) and downdrafts (-1 m s$^{-1}$ green isoline) in front of Liguria coastlines at 11:30 UTC from the WRF-IFS-DA simulation. The wind gust at 10-m above ground field is also shown in shaded colors.

**Supplementary references**

Charba, J.: Application of gravity current model to analysis of squall-line gust front, Mon. Wea. Rev., 102(2), 140–156, doi:10.1175/1520-0493(1974)102<0140:AOGCMT>2.0.CO;2, 1974.

Drobinski, P., Ducrocq, V., Alpert, P., Anagnostou, E., Béranger, K., Borga, M., Braud, I., Chanzy, A., Davolio, S., Delrieu, G., Estournel, C., Boubrahmi, N. F., Font, J., Grubišić, V., Gualdi, S., Homar, V., Ivančan-Picek, B., Kottmeier, C., Kotroni, V., Lagouvardos, K., Lionello, P., Llasat, M. C., Ludwig, W., Lutoff, C., Mariotti, A., Richard, E., Romero, R., Rotunno, R., Roussot, O., Ruin, I., Somot, S., Taupier-Letage, I., Tintore, J., Uijlenhoet, R. and Wernli, H.: HyMeX: A 10-year multidisciplinary program on the MediterraneanwWater cycle, Bull. Amer. Meteor. Soc., 95(7), 1063–1082, doi:10.1175/BAMS-D-12-00242.1, 2013.

Goff, R. C.: Vertical structure of thunderstorm outflows, Mon. Wea. Rev., 104(11), 1429–1440, doi:10.1175/1520-0493(1976)104<1429:VSOTO>2.0.CO;2, 1976.

Gonzalez, R. C. and Woods, R. E.: Digital Image Processing, 2nd edition, Prentice Hall, Upper Saddle River, N.J., 2002.

Grasso, L. D. (2000). The differentiation between grid spacing and resolution and their application to numerical modeling. Bull. Amer. Meteor. Soc., 81(3), 579-580.

Markowski, P. and Richardson, Y.: Mesoscale Meteorology in Mid-Latitudes, John Wiley & Sons, Ltd, Chichester, United Kingdom., 2010.

Mueller, C. K. and Carbone, R. E.: Dynamics of a thunderstorm outflow, J. Atmos. Sci., 44(15), 1879–1898, doi:10.1175/1520-0469(1987)044<1879:DOATO>2.0.CO;2, 1987.

Wakimoto, R. M.: The life cycle of thunderstorm gust fronts as viewed with Doppler radar and rawinsonde data, Mon. Wea. Rev., 110(8), 1060–1082, doi:10.1175/1520-0493(1982)110<1060:TLCOTG>2.0.CO;2, 1982.

---

## Author Comment (AC3) · 1 Mar 2020

**Answers to Reviewer #2**

**Summary and overview**: The paper describes the weather conditions in Genoa on the morning of 14 August 2018, when the collapse of a highway bridge resulted in numerous fatalities. Images captured by a security camera close to the bridge show strong winds and lightning during the collapse, which raises the question of a possible contribution of meteorological factors. Based on a diversity of remote sensing observations, including satellite, radar, lightning sensors and Doppler lidar, as well as a dense local network of surface stations, the paper highlights the presence of a thunderstorm located along the coast and of winds reaching 15 m/s due to the associated outflow. The event is hardly captured by convection-permitting WRF simulations due to its complex and local dynamics.

The paper describes a convective event based on comprehensive observations from a variety of sources. It presents interesting new material and is well written overall. However, it suffers several major flaws in its present form. The scientific topic of the paper is unclear and the main conclusions are not sufficiently supported. Perhaps related to the first issue, the discussion often spreads in diverging directions instead of focusing on relevant content, which renders the interpretation confusing or even speculative. In particular, the section based on model simulations is not convincing. General and specific comments are listed below to help alleviating these flaws. The paper thus requires substantial revision before it can be considered for publication in Natural Hazards and Earth System Sciences.

**Answer**: The reviewer's summary of our study in the first paragraph is accurate.

The overview provided in the second paragraph criticizes the scientific objective of this work and insufficient support of some of the conclusions. The model performances are also pointed out as being suspicious. We would like to emphasize that we have addressed all of these specific points below in the reviewer's general and specific comments. In the nutshell, the scientific objectives are strengthened, numerical simulations are additionally discussed and some of the conclusions will be reformulated in the revised manuscript. We thank the reviewer for the insightful comments provided in this review.

General comments:

**Q1**: What is the scientific topic of the paper? A hypothetical contribution of meteorological factors to the bridge collapse would involve engineering considerations and goes far beyond the scope of the paper. Furthermore, the results rather suggest that the wind was unexceptional, which is not clearly stated in the abstract and conclusions. The meteorological questions must be better introduced and the general knowledge that is acquired from that specific case study must be better highlighted.

**A1**: The objective of this study is to investigate the weather conditions during the collapse of the Morandi Bridge from different spatiotemporal scales. Besides inspecting the weather at local scale—namely above Genoa and around the collapsed bridge—the paper also analyzes the larger scale weather scenarios in order to identify the main contributors of this high impact weather, and to which extent these contributors were predictable in an operational setup of the Weather Research and Forecasting (WRF) model. The authors are convinced that this is a valid scientific objective because the paper does not only presents measurement data, but rather also intends to interpret and provide the physical background of the presented observations whenever possible.

Our objectives will be additionally strengthened in the abstract as well as the conclusions. Although wind speeds were much higher in comparison to the annual average value in Genoa, we agree with the reviewer

that the wind speed was not exceptionally high in terms of wind speed magnitude. Nevertheless, it was a thunderstorm event that occurred during the collapse and the unique lidar and other surface measurements deserve to be published and interpreted in order to provide a more complete picture of the circumstances surrounding the bridge collapse.

Lastly, we fully agree that a structural wind engineering study is beyond the scope of this paper, as well as this journal and that is why we opted not to present this analysis here. It might be carried out in the future research. We have also elaborated on this topic (i.e., the inclusion of wind engineering study) in our response to the general comment of Reviewer #1.

**Q2**: The interpretation of meteorological data, albeit interesting overall, tends to be confusing and sometimes speculative. In my opinion, the results suggest that two different gust fronts reached the western and eastern stations. This interpretation may be erroneous but the description of results and their presentation, e.g. the different space and time coordinates used for the different types of observations, prevents a clear picture of what happened. The analysis must be improved by emphasizing important information on figures, better connecting the different types of data and avoiding over interpretation.

**A2**: We thank the reviewer for stating that the overall interpretation of meteorological data is interesting.

We have now tried to remove all the speculations that were present in the original manuscript. In particular, some parts related to the interpretation of gust front structure from lidar measurements were now omitted (e.g., see also our answer to the question Q36 of Reviewer #1) and citing literature in the Introduction is limited to the most relevant studies for this research.

Our interpretation of data does not indicate the existence of two different gust fronts in the region. We are assuming that the reviewer is referring here to Figure 11 and the specific comment Q19 below. Different temporal signature of the gust front at different anemometer stations does not indicate that the event was characterized with several gust fronts, but rather a single gust front that evolved over space and time. The examples of this are numerous in literature. For instance, Burlando et al. (2017) analyzed a downburst in Livorno, Italy, on 1 October 2014 using a network of three anemometers installed along the coastline (two of them) and further in Livorno (one anemometer). While all three anemometers recorded the same event, the velocity (speed and direction) time histories were different at each anemometer. The reasons for different wind velocity signatures at different locations in the outflow are numerous, but some of the most obvious are that the outflow could be not perfectly symmetric in respect to downdraft touchdown due to the background atmospheric boundary layer winds and cloud translation, different surface roughness, as well as different radial distances of anemometers from the downburst center which results in naturally different evolution of the outflow at each anemometer.

One of the goals of this manuscript is to make the use of multiple data sources in order to describe and interpret weather conditions during the bridge collapse. Therefore, we have attempted to make multiple connections between different measurements throughout the manuscript. The radar reflectivity and precipitation data are linked, as well as the radar and satellite observations. Furthermore, we have connected the interpretation of lightning data with radar observations of precipitation zone. The high-frequency anemometer data positioned along the coastline are also interpreted in conjunction with Fig. 9, which shows the meteorological observations from the Genoa Airport weather station. The evolution of the precipitation zone and gust front are interpreted using multiple surface observations in Figures 10

and 11, respectively. Furthermore, the lidar measurements are discussed in terms of their relationship with the location of cloud cells and precipitation zone shown in Figures 6 and 7, respectively. Of course, the WRF numerical simulations were compared against the available in-situ and remote sensing measurements. However, we have additionally strengthened this aspect of the manuscript and additionally discussed the discrepancy between lightning data from LAMPINET and the video observation of lightning strike hitting the bridge prior to the collapse. Furthermore, the reasons for choosing the Genoa Airport weather station for determining the gust front displacement velocity instead of other weather stations will be also discussed in the revised manuscript. Also, the location of the precipitation zone in Figure 7 is now linked with the absence of lidar data in Figure 15.

**Q3**: The configuration of model simulations does not look appropriate for the event and their contribution to understanding its dynamics is very limited. Either run new model simulations and analyze results in details, or remove the section altogether

**A3**: The WRF configuration set chosen is the same of the operational model runs for this region by CIMA Research Foundation since the end of 2018 (https://www.cimafoundation.org/foundations/research-development/wrf.html). Furthermore, the schemes and model dynamics described in Section 2.2 demonstrate that we did not use any untested configuration of WRF. We fully agree with the reviewer that the model results presented in the previous version of the manuscript do not provide a good forecast of this event. In comparison to the original manuscript version, we will now add to the discussion the simulation including the data assimilation on the IFS driven forecast, namely WRF-IFS-DA. This prediction in comparison to the WRF-IFS experiment shows two main advantages: (1) wind peaks up to 12-13 m/s over the 8 wind stations used in the study, and (2) simulated event is one hour closer to the observed wind maxima (in comparison to the WRF-IFS experiment). Those improved wind forecast performances are due to the better reproduction by the WRF-IFS-DA simulation of the observed thunderstorm activity (Figure A below) close to Genoa in the late morning of 14 August 2018. The WRF-IFS-DA storms in the period 11–12 UTC resulted in a maximum wind gust around 15–20 m/s (Figure B below) and are located only 8–10 km far apart the wind stations locations. The estimated location error of the predicted storms is just 4–6 times the grid spacing. This model accuracy is deemed as very good (Grasso 2000).

[Figure]

Figure A. VMI map from WRF-IFS-DA simulation at 11:30 UTC.

[Figure]

Figure B. Maximum 5-min wind speeds at 10 m above ground from the WRF-IFS-DA simulation at the locations of eight anemometers situated in the Port of Genoa. See manuscript for additional information on anemometers location.

Lastly, we strongly believe that demonstrating that only one of the members of operational WRF suite properly capture this event is as scientifically significant as if the WRF provided a good comparison against

the observations in all cases. It is a truth that the model simulations are not able to perfectly reproduce the timing and location of the observed event. However, these simulations are performed in an operational framework and the location and timing error is in line with the state-of-the-art numerical weather modelling prediction. Furthermore, even with slightly different timing and location, the WRF-IFS-DA simulation is reproducing the event well enough to provide a deeper insight into the physics of this event.

Specific comments:

**Q4**: Abstract l. 25-26 Operational predictability cannot be discussed without using operational forecasts.

**A4**: As mentioned in A3, all the setups used in this study are operationally run by CIMA Research Foundation with the GFS initialization since the end of 2018 (https://www.cimafoundation.org/foundations/research-development/wrf.html) on behalf of ARPAL (Liguria Region Environment Protection Agency). The main aim of this section was, in fact, to investigate the predictability of this thunderstorm event in the operational weather forecasting model setups.

**Q5**: Introduction l. 28-33 Beginning with a discussion of bridge design is not appropriate in a geoscience journal.

**A5**: We decided to start the Introduction by introducing the reader with some basic information about the Morandi Bridge. We do not consider this to be a discussion on bridge design, but a very brief overview of the Morandi Bridge history and its state prior to the collapse. After all, the paper is analyzing the weather conditions during the collapse of this bridge, so it is appropriate to provide the basic information about the structure. However, we will remove the two sentences describing the bridge design in this paragraph, as the reviewer suggested.

We agree with the reviewer that the journal is not suited for structural analyses and technical terms in this field. That is why we have omitted this type of analysis in this manuscript (see also our answer A1 above).

**Q6**: l. 33-35, 117-122 Please clarify; I understand you want to be careful but the message is confusing as it is.

**A6**: As we stated in our answer A1, the goal of the manuscript is to document and investigate weather conditions during the collapse of the Morandi Bridge. The motivation for this meteorological study is the fact that the bridge collapse occurred during a severe thunderstorm event that was captured by several in-situ and remote measuring instruments. Besides only presenting the data, the study also aims to interpret the observations and provide the weather contributors for this event at different spatiotemporal scales.

However, we also want to emphasize that this study does not intend to explicitly demonstrate that the collapse occurred (or did not occur) due to the high impact weather. As we elaborated above, as well as in our answer to the general comment of Reviewer #1, a comprehensive structural wind engineering study would have to be conducted in order to quantify the likelihood of bridge collapse due to wind (or lightning). However, this type of study is beyond the scope of this paper and we are happy to notice that the reviewer agrees with us on this matter (Reviewer #2 question Q1 above).

**Q7**: l. 49-56 These claims need references.

**A7**: All this paragraph is based on newspaper articles (in Italian). A collection of these articles can be found in Wikipedia, see the Notes of the webpage https://it.wikipedia.org/wiki/Viadotto_Polcevera

**Q8**: l. 57-68 The paragraph does not fit with the topic: thunderstorms are not usually associated with cyclones in the mid latitudes; this may be different in the Mediterranean area but needs justification; the cited Zolt et al. (2006) does not mention downbursts.

**A8**: Cyclonic troughs are typically associated with frontal lines, which in turn can produce thunderstorms. We agree that this situation does not always occur, but the Alps orographic cyclogenesis that manifests around Genoa is usually associated with thunderstorms and downbursts (Burlando et al., 2017). The study of Zolt et al. (2006) as well as Burlando et al. (2018) further confirm this observation. We have accordingly modified our sentence in the revised manuscript to reflect the above clarification.

Zolt et al. (2006) describes the severe thunderstorm over Italy and while there is no explicit mention of a downburst, thunderstorms are inherently associated with downdrafts in the precipitation zone. These downdrafts further produce the radially advancing outflow known as the gust front. Furthermore, our citation of Zolt et al. (2006) is not related to the phenomena of downburst, but it is required in order to present other high impact weather thunderstorms that were investigated in this region. Conceptually, their study is somewhat similar to our paper and it is also published in the same journal.

However, kindly note that this paragraph in the revised manuscript will be drastically reduced and the discussion related to Zolt et al. (2006) shortened. We have limited the content to the most relevant studies and findings.

**Q9**: l. 78-90 The list of publications does not need to be exhaustive and must be shortened; the discussion of the number of Google Scholar publications is not relevant to motivate the study.

**A9**: We thank both reviewers for this suggestion. This part of the manuscript will be significantly shortened. Most of the citations are now excluded and the Google Scholar segment is also removed.

**Q10**: l. 99-105 It would be more logical to introduce these historical papers first, then the more recent examples, and finally the systematic bibliography above.

**A10**: This section will be also drastically shortened in accordance with Reviewer #1 comments. The segment related to aircraft accidents is removed from the revised manuscript.

**Q11**: 2. Data and Numerical Simulations l. 181-191 Why use two very different domains? The first one looks unnecessarily large for a thunderstorm case study.

**A11**: The domains in this study are the same as the domains used in the operational WRF forecasting for Liguria Region and the north-western Italy. The main purpose of the numerical simulations that we conducted is to investigate how good the operational WRF predicted this event. We agree that, being the focus a single thunderstorm event, a such larger domain would be unnecessary. Also see A3 and A4 for the details.

**Q12**: 3. Results and discussion: observations l. 217-230 The two low pressure systems over northern Europe are not relevant for the study (and the dates do not match); better focus on the region of interest and emphasize contributing factors, e.g., instability, cyclogenesis and fronts.

**A12**: This paragraph is a very short summary of the macro-meteorological conditions that ultimately led to the formation of the thunderstorm cloud on 14 August 2018 over Genoa city. Macro-meteorological conditions are not irrelevant for the local analysis and every meteorological office starts from analyzing the weather conditions at the synoptic scale and then focus on the local scale, because commonly the local weather is mainly due to the larger-scale conditions. In this case, the trough of Roswitha (see Fig. 4) caused a secondary cyclogenesis over the Gulf of Lion, which in turn was responsible of the atmospheric instability that led to the thunderstorm under study. As far as the dates mentioned are concerned, the lifetime of extratropical cyclones is usually in the order of some days: Roswitha was born on 16 August, but it actually affected the northern Mediterranean and Italy only a few days later.

As explained in A8, apart from single-cell thunderstorms which occur randomly during fair weather conditions, typically in summer when synoptic high-pressure systems prevent organized thunderstorm systems to develop and only thermally-driven cumulonimbus clouds generate thunderstorms, most often thunderstorms are related to unstable conditions linked to larger-scale disturbances, like troughs, cut-offs, or frontal areas associated to primary or secondary cyclones. In this paragraph, which is actually already very short, we describe the link between the "Weather conditions at larger scales" (title of Section 3.1) to the thunderstorm that is described later, also in Section 3.2 in terms of local observations. We cannot just start the description stating that on 14 August there were some clouds without explaining the reason behind the instability that triggered their formation.

**Q13**: l. 231-239 I do not clearly see the formation of a convective line; changing the color bar (16 km is reached in the tropics only) and improving the overall poor quality of Figure 5 may help.

**A13**: While the color bar goes up to 16 km, we state in that paragraph that the cloud tops above Genoa did not exceeded about 12 km. The quality of Figure 5 is now improved, but the color bar stayed the same because some of the clouds in the bottom-right in Figure 5a,b reached almost 16 km. Kindly note that we have improved Figure 5c with the zoom-in of the cloud tops above Genoa (also see our answers A15 and A16 to Reviewer #1 in relation to Figure 5).

The convection that stretches in the north-south direction above Genoa (Figure 5) resembles the main features of a convective line. The cloud tops are organized along the line that is very well structured. In addition, the convection in this region around 10:00 UTC is located below the tropopause height cutoff around 12:00 UTC (Figure 4b). This also reinforce the link between synoptic-scale conditions (Fig.4) and meso-scale weather (Fig. 5).

**Q14**: l. 243-257 This discussion appears speculative.

**A14**: As we stated above, this manuscript intends to go beyond the simple presentation of observational data and tries to interpret the observations using some physical arguments, whenever possible. We do agree that some of the statements might appear as speculative, but they are based on the observational evidence presented in data. Indeed, some other processes might have contributed and govern the cloud propagation, but the orographic influence that we "speculated" in this paragraph is also credible. Some relevant studies are also cited.

However, we will address this comment by relaxing our claim that orography likely influenced the cloud propagation. In this way we convey the uncertainty with this claim. In addition, please see our answer A17 to Reviewer #1 because it is also related to this discussion.

**Q15**: l. 277-278 This period includes the collapse time and should thus be shown and discussed.

**A15**: Figure 8c shows the number and spatial distribution of lightning strikes between 09:15 and 09:45 UTC, which encompasses the bridge collapse time. This discussion is provided later in the same paragraph. We observe that the lightning measuring network did not detect any strikes around the bridge in this period.

**Q16**: l. 284-310 Figure 9 needs improvement for interpretation: sampling of "20–30 min" and "approximately 2.5 hours" are confusing and time labels every 2:24 h are weird; better zoom on the time of interest and show every single point of measurement, including a time series of wind gusts rather than a single value; the observed impact on pressure is very speculative and is better omitted; finally, the temperature and wind change occurs over a period of about 1 h, which is not "abrupt" and does not support the presence of a macroburst.

**A16**: This figure is now improved and considerably modified. Firstly, we have used symbols instead of line for wind direction, and symbols were also added to air pressure and air temperature lines. This edit was also proposed by Reviewer #1 in the comment Q23.

The measurements are available in a non-standard hours (i.e., not in the full hour) and that is why the time labels are not full hour. However, we have modified the time labels to a full hour and not 24 min into an hour in order to facilitate this comment (2-h interval shown in the *x*-axis). However, kindly note that not all measurements are available in the full hour timestamp.

There is not time series of wind gust, but only this single observed gust that is shown in Figure 9a. The gusts are recorded if the wind characteristics satisfy given conditions over a given period of time. The only significant gust recorded at the weather station was the one shown in Figure 9a that occurred around the bridge collapse time.

Unfortunately, the time resolution of air pressure data is much lower than the air temperature and wind velocity. Despite this, however, we still observe an overall positive trend in the air pressure which is in accordance with thunderstorm passage. Moreover, the pressure increase associated with thunderstorm passage is more gradual than wind speed and temperature jumps (e.g., Markowski and Richardson, 2010). We fully agree with the reviewer that the high-frequency time history of surface air pressures would probably deviate from the data shown in Figure 9c, but unfortunately such data are not available. However, we have reformulated our statements in the revised manuscript and excluded the terms "rapid" and "abrupt" when talking about air temperature and pressures. The kindly disagree with the reviewer's statement that the data do not resemble a downburst-like signature. The velocity and air temperature data alone do indicate a downburst passage over the area.

**Q17**: l. 305-308 This belongs to the introduction.

**A17**: This part was included in order to demonstrate that the observational data fit some conceptual models of thunderstorm downbursts. However, we will remove it from the revised manuscript. The proper description of downbursts and gust fronts will be included in introduction, which is also in accordance with the reviewer's comment Q20 below.

**Q18**: l. 311-314 The difference between coastal and terrain station is more striking than between west and east; drop symbols are illustrative but not quantitative, numbers are also needed.

**A18**: We will include numbers next to drop symbols and station labels (the units are mm h$^{-1}$). However, it still seems that the increase of precipitation along the coast (towards the east) is more pronounced than moving further inland from the coast. However, we also notice that the west-east span of stations is approximately two times larger than the south-north span, which could bias these conclusions. We will add this statement in the revised manuscript.

**Q19**: l. 319-341 The interpretation is not convincing: eastern and western stations clearly behave differently, which is consistent with the location of the convective cell over the eastern stations; however, the slow wind turning at western stations does not support the presence of a macroburst; moreover, the claimed association with a gust front is confusing without a spatial representation; what about temperature records at nearby stations?

**A19**: This part of the paper was not clear to Reviewer #1 as well in **Q26**, which means for sure that the analysis that we have described based on wind speed and direction data is not clear enough. The point is that the time series have a behavior which is absolutely coherent with the passage of a gust front that is spreading northward (away from the cloud downdraft) but, in turn, is also travelling northeastward together with the cloud itself. Because this process might not easy to visualize for the reader, this part has to be properly clarified. In the revised paper, new pictures/panels will be added to Fig. 11 (not to increase the overall number of figures) with the wind plotted as arrows and the text will be changed accordingly.

**Q20**: l. 341-346, 372-374 This belongs to the introduction; a definition of downburst, macroburst and gust front is lacking and associated cold pools should be explained as well.

**A20**: The downburst definition and origins are already provided in the Introduction. However, we will additionally address this comment and also include the definitions of dry and wet downbursts, microbursts and macrobursts, as well as the thunderstorm gust front in the new version of this manuscript.

**Q21**: l. 356-371 This discussion is lengthy and should be streamlined; zooming in would help identifying features on Figure 12, which is white mostly (no data).

**A21**: Figure 12 is now substantially improved (also see A29 in rebuttal for Reviewer #1). Namely, the color bar is restricted to -7 m s$^{-1}$ to +17 m s$^{-1}$, which is the range of radial velocities captured during this event. Secondly, the figure is saved with much higher resolution (600 dpi). Thirdly, the radial distance is now limited to 5492 m away from the lidar because the data beyond that radius are either unreliable or non-existent due to precipitation (i.e., mostly white as the reviewer correctly noticed).

**Q22**: l. 374-385 (and 422-424) The eight symbols in Figure 13 depict eight different times, thus their representation is confusing and the computation of a displacement velocity obscure (and of the inclination angle).

**A22**: In addition to displacement velocity calculation that is provided in the manuscript, we kindly direct the reviewer to our answer A31 to Reviewer #1. We have clarified how the gust front height and displacement velocities were calculated including the treatment of lidar scanning time and the time needed to move to the next elevation. This lidar makes one elevation scan in 49 s and it takes 2 s to move to the next elevation. For example, this scanning velocity is much higher than that of a Doppler radar. Yet,

in almost all practical analyses, the weather radar data (i.e., volume scans) are considered to be instantaneous despite the fact that the radar actually need some finite amount of time to perform the scan.

Here, an analogy can be drawn to rainfall data in order to further clarify this comment. Namely, concerning radar rainfall, it is usually assumed that the rainfall estimates from the "instantaneous" reflectivity represent the average rainfall in the interval 5–10 minutes around the scan. This is why in about 5-10 minutes (time taken for one complete scan, depending on the radar characteristics) the decorrelation in time is reasonably low and an instantaneous observation in space, at the resolution of the radar, taken any time inside the 5 minutes can be representative of the average rain rate in a 1 km x 1 km x 5 (10) min cell (assuming the radar resolution of 1 km). The similar assumption of the "frozen flow field" over the lidar scan period was invoked in the lidar wind analyses.

**Q23**: l. 386-429 Considering the small contribution of moisture, the uncertainty in k and the very speculative increase in pressure, the computation and discussion should be largely simplified.

**A23**: While the reviewer might disagree, the authors believe that this discussion is simplistic. Namely, the authors' goal is to be as transparent as possible and to explain the reader how the displacement velocity was obtained. The objective is to show the method that can be easily replicated if the reader is interested. That is why we have included all the steps and explained them—however, the steps are arguably quite simple.

The uncertainty of k and low temporal resolution of pressure data are highlighted in order to inform the reader that the analysis contains uncertainties. The Genoa Airport station was used due to its proximity to the sea (i.e., lidar is also located on the coastline). We have highlighted this in the revised paper. Furthermore, our results are quite similar to the results reported in Goff (1976) and Mueller and Carbone (1987), but also very different to the results presented in Charba (1974). We believe that this comparison to other literature is very important in order to justify our findings, but also to demonstrate there is a large discrepancy between different papers. Both information are valuable. In addition, we have emphasized now that the reported displacement velocity is only the projection of the overall displacement velocity vector in the direction of lidar. Kindly also see our answer A32 to Reviewer #1 due to its similarity with this comment.

The smaller influence of relative humidity to the results is not limited to our data, but to the physic of this phenomenon. This quantity is less important factor in the gust front propagation velocity and dynamics, and we wanted to demonstrate that in our analysis. We believe that such an analysis is very instructive for the readers.

**Q24**: l. 430-455 This discussion is largely speculative and should be streamlined or omitted.

**A24**: We agree that this discussion might seem speculative in some parts, but in reality it is steered by the analytical results that were obtained using observational data and/or published literature. This is the authors' interpretation of observational evidences and we agree that perhaps some other interpretations would be possible due to the lack of higher-quality data that would provide the clearer picture of this event.

We would like to emphasize once more that the authors believe that the additional value of this research is in its attempt to provide physical background for the presented observational data. Otherwise, the

paper becomes a technical report and this is not our intent. However, we will address this comment by excluding the most speculative parts in this section (e.g., the segment related to the variability of spatial separation between the high-speed regions in Figure 12).

**Q25**: l. 456-463 What can be learned from Figure 15?

**A25**: There are few reasons for the inclusion of Figure 15 in this manuscript. Firstly, we wanted to present to the readers that the lidar did not observe any significant phenomena during the bridge collapse due to the precipitation zone that was located close to the lidar. The overall goal here is to present what was the weather like during the collapse and it is important to document that this lidar could not provide a significant insight into it.

Secondly, the limited data that are available from the lidar show that the easterly winds were present at the lidar locations during the bridge collapse. This result is in agreement with the location of precipitation zone shown in Figure 7. This observation is also in accordance with the reviewer's comment Q2 that recommends more connection between different data sources in the interpretation of results. Kindly note that this figure is significantly improved in the revised manuscript in terms of color bar limits ($-7$ m s$^{-1}$ to 9 m s$^{-1}$) and the radial distance of data is limited to 2852 m away from the lidar.

**Q26**: 4. Results and discussion: WRF Numerical Simulations l. 465-517 The purpose of this section is unclear: even the best run (WRF-IFS) still strongly differs from observations thus would need a much more detailed and systematic analysis to provide useful information on the actual dynamics of the event, while the model configuration is too different in the other two runs to provide useful information about model sensitivity with such a local event.

**A26**: The revised manuscript contains new numerical simulations. We have added WRF-IFS-DA run to supplement previous three simulations. This newest simulation provides much better agreement with the observations. While the onset of this thunderstorm is still delayed in the model, the delay is significantly smaller than in the previous simulations.

The purpose of this section is to analyze the accuracy and reliability of operational WRF forecasts in the case of this particular event. The authors are convinced that "imperfect" results from the operational weather forecast are very valuable addition to this manuscript because they demonstrate that the event could not have been accurately predicted on the morning of the disaster. In fact, we believe that the slightly negative results from the numerical model are more valuable than if the model perfectly predicted this event.

However, note that the new version of this manuscript contains additional WRF outputs that are now used to describe thunderstorm dynamics, microphysics processes (including the spatial distribution of water species). For instance, the figure below shows the vertical cross-section of the WRF-IFS-DA simulated radar reflectivity along the convective cell that developed in the Genoa region at 11:30 UTC. The shaded colors in the horizontal plan show the air temperature at 2 m above ground. As we mentioned earlier, the WRF still delayed this thunderstorm for approximately 2 hours. The revised manuscript contain discussions that describe this and other figures from the new WRF simulations.

To conclude, we now added to the discussion the simulation that includes the data assimilation on the IFS driven forecast, namely WRF-IFS-DA, to complete the simulations set. The aim of the numerical simulations was to analyze the behavior of operational setups two different initialization. As a result, the

IFS and GFS gave different results and the IFS simulations resulted in a good reproduction of the event especially when the DA is used. The use of an ensemble of simulations at very high resolution using different microphysics and dynamical setup is beyond the scope of this paper. However, recently Parodi et al. (2019) analyzed a macroburst that took place over the same region on 14 October 2016. Their analysis employed an ensemble of kilometer-scale simulations using different microphysics and planetary boundary layer schemes. Despite this, still only few members performed well and those are the one that used a setup similar to the one used in the operational forecast that were replicated in this study. Finally, the revised version of the paper will contain a deeper analysis on the best performing simulation, and the paper by Parodi et al. (2019) will be cited.

**Q27**: 5. Conclusions l. 551-556 The study clearly shows the presence of a thunderstorm during the storm collapse but rather suggests that associated winds were not extreme; how these may have or not have affected the bridge is far beyond the scope of the study.

**A27**: As already stated, a detailed investigation of the causes of bridge collapse from the engineering point of view is beyond the scope of this paper and this journal. However, it is also true that at this moment, after 1.5 years from the collapse, the cause of this disaster is still not clear. The structure was surely old and not well maintained, therefore it is possible that we are not facing a disaster due to an extreme and exceptional meteorological event, but rather a collapse due to a strong, even if not extreme, weather condition. It might be, therefore, that the common paradigm that a disaster caused by a natural hazard can be triggered only by an extreme event, has to be widened to include non-extreme events when considering elder and weaker structures.

**Supplementary references**

Burlando, M., Romanić, D., Solari, G., Hangan, H. and Zhang, S.: Field data analysis and weather scenario of a downburst event in Livorno, Italy, on 1 October 2012, Mon. Wea. Rev., 145(9), 3507–3527, doi:10.1175/MWR-D-17-0018.1, 2017.

Burlando, M., Zhang, S. and Solari, G.: Monitoring, cataloguing, and weather scenarios of thunderstorm outflows in the northern Mediterranean, Nat. Hazards Earth Syst. Sci., 18(9), 2309–2330, doi:https://doi.org/10.5194/nhess-18-2309-2018, 2018.

Charba, J.: Application of gravity current model to analysis of squall-line gust front, Mon. Wea. Rev., 102(2), 140–156, doi:10.1175/1520-0493(1974)102<0140:AOGCMT>2.0.CO;2, 1974.

Goff, R. C.: Vertical structure of thunderstorm outflows, Mon. Wea. Rev., 104(11), 1429–1440, doi:10.1175/1520-0493(1976)104<1429:VSOTO>2.0.CO;2, 1976.

Grasso, L. D. (2000). The differentiation between grid spacing and resolution and their application to numerical modeling. Bull. Amer. Meteor. Soc., 81(3), 579-580.

Markowski, P. and Richardson, Y.: Mesoscale Meteorology in Mid-Latitudes, John Wiley & Sons, Ltd, Chichester, United Kingdom., 2010.

Mueller, C. K. and Carbone, R. E.: Dynamics of a thunderstorm outflow, J. Atmos. Sci., 44(15), 1879–1898, doi:10.1175/1520-0469(1987)044<1879:DOATO>2.0.CO;2, 1987.

Parodi, A., Lagasio, M., Maugeri, M., Turato, B., & Gallus, W. (2019). Observational and modelling study of a major downburst event in Liguria: The 14 October 2016 case. Atmosphere, 10(12), 788.

Zolt, S. D., Lionello, P., Nuhu, A. and Tomasin, A.: The disastrous storm of 4 November 1966 on Italy, Nat. Hazards Earth Syst. Sci., 6(5), 861–879, doi:https://doi.org/10.5194/nhess-6-861-2006, 2006.